# Fool SHAP with Stealthily Biased Sampling.

**Gabriel Laberge**[1], **Ulrich Aïvodji**[2], **Satoshi Hara**[3], **Mario Marchand**[4], **Foutse Khomh**[1]

[1]Polytechnique Montréal, Québec [2]École de technologie supérieure, Québec
[3]Osaka University, Japan [4]Universitié de Laval à Québec

`{gabriel.laberge,foutse.khomh}@polymtl.ca`
`ulrich.aivodji@etsmtl.ca`
`satohara@ar.sanken.osaka-u.ac.jp`
`mario.marchand@ift.ulaval.ca`

## Abstract

SHAP explanations aim at identifying which features contribute the most to the difference in model prediction at a specific input versus a background distribution. Recent studies have shown that they can be manipulated by malicious adversaries to produce arbitrary desired explanations. However, existing attacks focus solely on altering the black-box model itself. In this paper, we propose a complementary family of attacks that leave the model intact and manipulate SHAP explanations using stealthily biased sampling of the data points used to approximate expectations w.r.t the background distribution. In the context of fairness audit, we show that our attack can reduce the importance of a sensitive feature when explaining the difference in outcomes between groups while remaining undetected. More precisely, experiments performed on real-world datasets showed that our attack could yield up to a 90% relative decrease in amplitude of the sensitive feature attribution. These results highlight the manipulability of SHAP explanations and encourage auditors to treat them with skepticism.

## 1 Introduction

As Machine Learning (ML) gets more and more ubiquitous in high-stake decision contexts (*e.g.* , healthcare, finance, and justice), concerns about its potential to lead to discriminatory models are becoming prominent. The use of auditing toolkits (Adebayo et al., 2016; Saleiro et al., 2018; Bellamy et al., 2018) is getting popular to circumvent the use of unfair models. However, although auditing toolkits can help model designers in promoting fairness, they can also be manipulated to mislead both the end-users and external auditors. For instance, a recent study of Fukuchi et al. (2020) has shown that malicious model designers can produce a benchmark dataset as fake "evidence" of the fairness of the model even though the model itself is unfair.

Another approach to assess the fairness of ML systems is to explain their outcome in a *post hoc* manner (Guidotti et al., 2018). For instance, SHAP (Lundberg & Lee, 2017) has risen in popularity as a means to extract model-agnostic local feature attributions. Feature attributions are meant to convey how much the model relies on certain features to make a decision at some specific input. The use of feature attributions for fairness auditing is desirable for cases where the interest is on the direct impact of the sensitive attributes on the output of the model. One such situation is in the context of causal fairness (Chikahara et al., 2021). In some practical cases, the outputs cannot be independent from the sensitive attribute unless we sacrifice much of prediction accuracy. For example, any decisions based on physical strength are statistically correlated to gender due to biological nature. The problem in such a situation is not the statistical bias (such as demographic parity), but whether the decision is based on physical strength or gender, *i.e.* the attributions of each feature.

The focus of this study is on manipulating the feature attributions so that the dependence on sensitive features is hidden and the audits are misled as if the model is fair even if it is not the case. Recently, several studies reported that such a manipulation is possible, *e.g.* by modifying the black-box model to be explained (Slack et al., 2020; Begley et al., 2020; Dimanov et al., 2020), by manipulating the computation algorithms of feature attributions (Aïvodji et al., 2019), and by poisoning the data distribution (Baniecki et al., 2021; Baniecki & Biecek, 2022). With these findings in mind, the current

possible advice to the auditors is not to rely solely on the reported feature attributions for fairness auditing. A question then arises about what "evidence" we can expect in addition to the feature attributions, and whether they can be valid "evidence" of fairness.

In this study, we show that we can craft fake "evidence" of fairness for SHAP explanations, which provides the first negative answer to the last question. In particular, we show that we can produce not only manipulated feature attributions but also a benchmark dataset as the fake "evidence" of fairness. The benchmark dataset ensures the external auditors reproduce the reported feature attributions using the existing SHAP library. In our study, we leverage the idea of stealthily biased sampling introduced by Fukuchi et al. (2020) to cherry-pick which data points to be included in the benchmark. Moreover, the use of stealthily biased sampling allows us to keep the manipulation undetected by making the distribution of the benchmark sufficiently close to the true data distribution. Figure 1 illustrates the impact of our attack in an explanation scenario with the Adult Income dataset.

Our contributions can be summarized as follows:

- Theoretically, we formalize a notion of foreground distribution that can be used to extend Local Shapley Values (LSV) to Global Shapley Values (GSV), which can be used to decompose fairness metrics among the features (Section 2.2). Moreover, we formalize the task of manipulating the GSV as a Minimum Cost Flow (MCF) problem (Section 4).

- Experimentally (Section 5), we illustrate the impact of the proposed manipulation attack on a synthetic dataset and four popular datasets, namely Adult Income, COMPAS, Marketing, and Communities. We observed that the proposed attack can reduce the importance of a sensitive feature while keeping the data manipulation undetected by the audit.

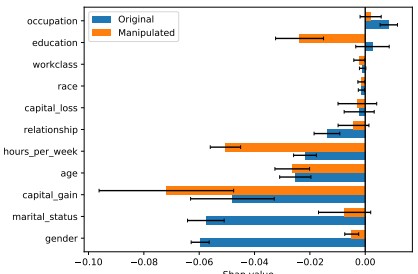

Figure 1: Example of our attack on the Adult Income dataset. After the attack, the feature gender moved from the most negative attribution to the $6^{th}$, hence hiding some of the model bias.

Our results indicate that SHAP explanations are not robust and can be manipulated when it comes to explaining the difference in outcomes between groups. Even worse, our results confirm we can craft a benchmark dataset so that the manipulated feature attributions are reproducible by external audits. Henceforth, we alert auditors to treat post-hoc explanation methods with skepticism even if it is accompanied by some additional evidence.

## 2 SHAPLEY VALUES

### 2.1 LOCAL SHAPLEY VALUES

Shapley values are omnipresent in post-hoc explainability because of their fundamental mathematical properties (Shapley, 1953) and their implementation in the popular SHAP Python library (Lundberg & Lee, 2017). SHAP provides local explanations in the form of feature attributions *i.e.* given an input of interest $\boldsymbol{x}$, SHAP returns a score $\phi_i \in \mathbb{R}$ for each feature $i = 1, 2, \ldots, d$. These scores are meant to convey how much the model $f$ relies on feature $i$ to make its decision $f(\boldsymbol{x})$. Shapley values have a long background in coalitional game theory, where multiple players collaborate toward a common outcome. In the context of explaining model decisions, the players are the input features and the common outcome is the model output $f(\boldsymbol{x})$. In coalitional games, players (features) are either present or absent. Since one cannot physically remove an input feature once the model has already been fitted, SHAP removes features by replacing them with a baseline value $\boldsymbol{z}$. This leads to the *Local Shapley Value* (LSV) $\phi_i(f, \boldsymbol{x}, \boldsymbol{z})$ which respect the so-called efficiency axiom (Lundberg & Lee, 2017)

$$\sum_{i=1}^{d} \phi_i(f, \boldsymbol{x}, \boldsymbol{z}) = f(\boldsymbol{x}) - f(\boldsymbol{z}). \tag{1}$$

Simply put, the difference between the model prediction at $\boldsymbol{x}$ and the baseline $\boldsymbol{z}$ is shared among the different features. Additional details on the computation of LSV are presented in Appendix B.1.

## 2.2 GLOBAL SHAPLEY VALUES

LSV are local because they explain the prediction at a specific $x$ and rely on a single baseline input $z$. Since model auditing requires a more global analysis of model behavior, we must understand the predictions at multiple inputs $x \sim \mathcal{F}$ sampled from a distribution $\mathcal{F}$ called the *foreground*. Moreover, because the choice of baseline is somewhat ambiguous, the baselines are sampled $z \sim \mathcal{B}$ from a distribution $\mathcal{B}$ colloquially referred to as the *background*. Taking inspiration from Begley et al. (2020), we can compute *Global Shapley Values* (GSV) by averaging LSV over both foreground and background distributions.

**Definition 2.1.**
$$\Phi_i(f, \mathcal{F}, \mathcal{B}) := \mathop{\mathbb{E}}_{\substack{x \sim \mathcal{F} \\ z \sim \mathcal{B}}} [\phi_i(f, x, z)], \quad i = 1, 2, \ldots, d. \tag{2}$$

**Proposition 2.1.** *The GSV have the following property*
$$\sum_{i=1}^{d} \Phi_i(f, \mathcal{F}, \mathcal{B}) = \mathop{\mathbb{E}}_{x \sim \mathcal{F}} [f(x)] - \mathop{\mathbb{E}}_{x \sim \mathcal{B}} [f(x)]. \tag{3}$$

## 2.3 MONTE-CARLO ESTIMATES

In practice, computing expectations w.r.t the whole background and foreground distributions may be prohibitive and hence Monte-Carlo estimates are used. For instance, when a dataset is used to represent a background distribution, explainers in the SHAP library such as the ExactExplainer and TreeExplainer will subsample this dataset [1] by selecting 100 instances uniformly at random when the size of the dataset exceeds 100. More formally, let

$$\mathcal{C}(S, \boldsymbol{\omega}) := \sum_{x^{(j)} \in S} \omega_j \delta(x^{(j)}) \tag{4}$$

represent a categorical distribution over a finite set of input examples $S$, where $\delta(\cdot)$ is the Dirac probability measure, $w_j \geqslant 0 \; \forall j$, and $\sum_j \omega_j = 1$. Estimating expectations with Monte-Carlo amounts to sampling $M$ instances

$$S_0 \sim \mathcal{F}^M \qquad S_1 \sim \mathcal{B}^M, \tag{5}$$

and computing the plug-in estimate

$$\begin{aligned} \widehat{\boldsymbol{\Phi}}(f, S_0, S_1) :=& \boldsymbol{\Phi}(f, \mathcal{C}(S_0, \mathbf{1}/M), \mathcal{C}(S_1, \mathbf{1}/M)) \\ =& \frac{1}{M^2} \sum_{x^{(i)} \in S_0} \sum_{z^{(j)} \in S_1} \boldsymbol{\phi}(f, x^{(i)}, z^{(j)}). \end{aligned} \tag{6}$$

When a set of samples is a singleton (*e.g.* $S_1 = \{z^{(j)}\}$), we shall use the convention $\widehat{\boldsymbol{\Phi}}(f, S_0, \{z^{(j)}\}) \equiv \widehat{\boldsymbol{\Phi}}(f, S_0, z^{(j)})$ to improve readability. In Appendix B.2, $\widehat{\boldsymbol{\Phi}}(f, S_0, S_1)$ is shown to be a consistent and asymptotically normal estimate of $\boldsymbol{\Phi}(f, \mathcal{F}, \mathcal{B})$ meaning that one can compute approximate confidence intervals around $\widehat{\boldsymbol{\Phi}}$ to capture $\boldsymbol{\Phi}$ with high probability. In practice, the estimates $\widehat{\boldsymbol{\Phi}}$ are employed as the model explanation which we see as a vulnerability. As discussed in Section 4, the Monte-Carlo estimation is the key ingredient that allows us to manipulate the GSV in favor of a dishonest entity.

## 3 AUDIT SCENARIO

This section introduces an audit scenario to which the proposed attack of SHAP can apply. This scenario involves two parties: a company and an audit. The company has a dataset $D = \{(x^{(i)}, y^{(i)})\}_{i=1}^{N}$ with $x^{(i)} \in \mathbb{R}^d$ and $y^{(i)} \in \{0, 1\}$ that contains $N$ input-target tuples and also has a model $f : \mathcal{X} \to [0, 1]$ that is meant to be deployed in society. The binary feature with index $s$ (*i.e.* $x_s \in \{0, 1\}$) represents a sensitive feature with respect to which the model should not explicitly

---
[1] https://github.com/slundberg/shap/blob/0662f4e9e6be38e658120079904899cccda59ff8/shap/maskers/_tabular.py#L54-L55

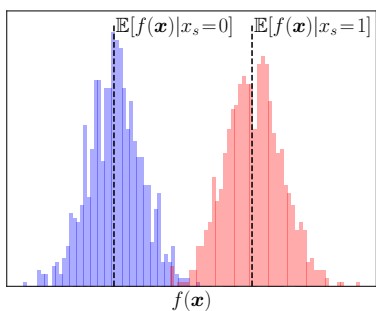 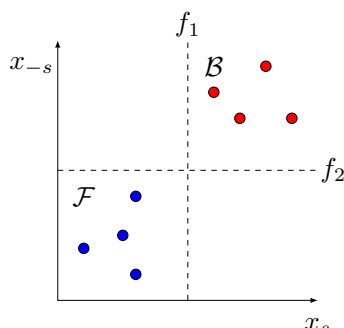

(a) The data initially provided by the company to the audit is $f(D_0)$ and $f(D_1)$ *i.e.* the model predictions for all instances in the private dataset for different values of $x_s$. This dataset can later be used by the audit to assess whether or not the subsets $S_0', S_1'$ provided by the company where cherry-picked.

(b) Two models $f_1$ and $f_2$ (decision boundaries in dashed lines) with perfect accuracy exhibit a disparity in outcomes w.r.t groups with $x_s < 0$ and $x_s > 0$. Here, $\Phi_s(f_1, \mathcal{F}, \mathcal{B}) = -1$ while $\Phi_s(f_2, \mathcal{F}, \mathcal{B}) = 0$. Hence, $f_2$ is **indirectly** unfair toward $x_s$ because of correlations in the data.

Figure 2: Illustrations of the audit scenario.

discriminate. Both the data $D$ and the model $f$ are highly private so the company is very careful when providing information about them to the audit. Hence, $f$ is a black box from the point of view of the audit. At first, the audit asks the company for the necessary data to compute fairness metrics *e.g.* the Demographic Parity (Dwork et al., 2012), the Predictive Equality (Corbett-Davies et al., 2017), or the Equal Opportunity (Hardt et al., 2016). Note that our attack would apply as long as the fairness metric is a difference in model expectations over subgroups. For simplicity, the audit decides to compute the Demographic Parity

$$\mathbb{E}[f(\boldsymbol{x})|x_s = 0] - \mathbb{E}[f(\boldsymbol{x})|x_s = 1], \qquad (7)$$

and therefore demands access to the model outputs for all inputs with different values of the sensitive feature : $f(D_0)$ and $f(D_1)$, where $D_0 = \{\boldsymbol{x}^{(i)} : x_s^{(i)} = 0\}$ and $D_1 = \{\boldsymbol{x}^{(i)} : x_s^{(i)} = 1\}$ are subsets of the input data of sizes $N_0$ and $N_1$ respectively. Doing so does not force the company to share values of features other than $x_s$ nor does it requires direct access to the inner workings of the proprietary model. Hence, this demand respects privacy requirements and the company will accept to share the model outputs across all instances, see Figure 2a. At this point, the audit confirms that the model is indeed biased in favor of $x_s = 1$ and puts in question the ability of the company to deploy such a model. Now, the company argues that, although the model exhibits a disparity in outcomes, it does not mean that the model explicitly uses the feature $x_s$ to make its decision. If such is the case, then the disparity could be explained by other features statistically associated with $x_s$. Some of these other features may be acceptable grounds for decisions. To verify such a claim, the audit decides to employ post-hoc techniques to explain the disparity. Since the model is a black-box, the audits shall compute the GSV. The foreground $\mathcal{F}$ and background $\mathcal{B}$ are chosen to be the data distributions conditioned on $x_s = 0$ and $x_s = 1$ respectively

$$\mathcal{F} := \mathcal{C}(D_0, \mathbf{1}/N_0) \qquad \mathcal{B} := \mathcal{C}(D_1, \mathbf{1}/N_1). \qquad (8)$$

According to Equation 3, the resulting GSV will sum up to the demographic parity (cf. Equation 7). If the sensitive feature has a large negative GSV $\Phi_s$, then this would mean that the model is **explicitly** relying on $x_s$ to make its decisions and the company would be forbidden from deploying the model. If the GSV has a small amplitude, however, the company could still argue in favor of deploying the model in spite of having disparate outcomes. Indeed, the difference in outcomes by the model could be attributed to more acceptable features. See Figure 2b for a toy example illustrating this reasoning.

To compute the GSV, the audit demands the two datasets of inputs $D_0$ and $D_1$, as well as the ability to query the black box $f$ at arbitrary points. Because of privacy concerns on sharing values of $\boldsymbol{x}$ across the whole dataset, and because GSV must be estimated with Monte-Carlo, both parties agree that the company shall only provide subsets $S_0 \subset D_0$ and $S_1 \subset D_1$ of size $M$ to the audit so they can compute a Monte-Carlo estimate $\hat{\boldsymbol{\Phi}}(f, S_0, S_1)$. The company first estimate GSV on their own by choosing $S_0, S_1$ uniformly at random from $\mathcal{F}$ and $\mathcal{B}$ (cf. Equation 5) and observe that $\hat{\Phi}_s$ indeed has a

large negative value. They realize they must carefully select which data points will be sent, otherwise, the audit may observe the bias toward $x_s = 1$ and the model will not be deployed. Moreover, the company understands that the audit currently has access to the data $f(D_0)$ and $f(D_1)$ representing the model predictions on the whole dataset (see Figure 2a). Therefore, if the company does not share subsets $S_0, S_1$ that were chosen uniformly at random from $D_0, D_1$, it is possible for the audit to detect this fraud by doing a statistical test comparing $f(S_0)$ to $f(D_0)$ and $f(S_1)$ to $f(D_1)$. The company needs a method to select **misleading subsets** $S_0', S_1'$ whose GSV is manipulated in their favor while remaining undetected by the audit. Such a method is the subject of the next section.

## 4 FOOL SHAP WITH STEALTHILY BIASED SAMPLING

### 4.1 MANIPULATION

To fool the audit, the company can decide to indeed sub-sample $S_0'$ uniformly at random $S_0' \sim \mathcal{F}^M$. Then, given this choice of foreground data, they can repeatedly sub-sample $S_1' \sim \mathcal{B}^M$, and choose the set $S_1'$ leading to the smallest $|\widehat{\Phi}_s(f, S_0', S_1')|$. We shall call this method "brute-force". Its issue is that, by sub-sampling $S_1'$ from $\mathcal{B}$, it will take an enormous number of repetitions to reduce the attribution since the GSV $\widehat{\Phi}_s(f, S_0', S_1')$ is concentrated on the population GSV $\Phi_s(f, \mathcal{F}, \mathcal{B})$.

A more clever method is to re-weight the background distribution before sampling from it *i.e.* define $\mathcal{B}_{\boldsymbol{\omega}}' := \mathcal{C}(D_1, \boldsymbol{\omega})$ with $\boldsymbol{\omega} \neq \mathbf{1}/N_1$ and then sub-sample $S_1' \sim \mathcal{B}_{\boldsymbol{\omega}}'^M$. To make the model look fairer, the company needs the $\widehat{\Phi}_s$ computed with these cherry-picked points to have a small magnitude.

**Proposition 4.1.** *Let $S_0'$ be **fixed**, and let $\xrightarrow{p}$ represent convergence in probability as the size $M$ of the set $S_1' \sim \mathcal{B}_{\boldsymbol{\omega}}'^M$ increases, we have*

$$\widehat{\Phi}_s(f, S_0', S_1') \xrightarrow{p} \sum_{\boldsymbol{z}^{(j)} \in D_1} \omega_j \, \widehat{\Phi}_s(f, S_0', \boldsymbol{z}^{(j)}). \tag{9}$$

We note that the coefficients $\widehat{\Phi}_s(f, S_0', \boldsymbol{z}^{(j)})$ in Equation 9 are tractable and can be computed and stored by the company. We discuss in more detail how to compute them in Appendix B.3. An additional requirement is that the non-uniform distribution $\mathcal{B}_{\boldsymbol{\omega}}'$ remains similar to the original $\mathcal{B}$. Otherwise, the fraud could be detected by the audit. In this work, the notion of similarity between distributions will be captured by the Wasserstein distance in output space.

**Definition 4.1** (Wassertein Distance). *Any probability measure $\pi$ over $D_1 \times D_1$ is called a coupling measure between $\mathcal{B}$ and $\mathcal{B}_{\boldsymbol{\omega}}'$, denoted $\pi \in \Delta(\mathcal{B}, \mathcal{B}_{\boldsymbol{\omega}}')$, if $1/N_1 = \sum_j \pi_{ij}$ and $\omega_j = \sum_i \pi_{ij}$. The Wassertein distance between $\mathcal{B}$ and $\mathcal{B}_{\boldsymbol{\omega}}'$ mapped to the output-space is defined as*

$$\mathcal{W}(\mathcal{B}, \mathcal{B}_{\boldsymbol{\omega}}') = \min_{\pi \in \Delta(\mathcal{B}, \mathcal{B}_{\boldsymbol{\omega}}')} \sum_{i,j} |f(\boldsymbol{z}^{(i)}) - f(\boldsymbol{z}^{(j)})| \pi_{ij}, \tag{10}$$

*a.k.a the cost of the optimal transport plan that distributes the mass from one distribution to the other.*

We propose Algorithm 1 to compute the weights $\boldsymbol{\omega}$ by minimizing the magnitude of the GSV while maintaining a small Wasserstein distance. The trade-off between attribution manipulation and proximity to the data is tuned via a hyper-parameter $\lambda > 0$. We show in the Appendix A.2 that the optimization problem at line 5 of Algorithm 1 can be reformulated as a Minimum Cost Flow (MCF) and hence can be solved in polynomial time (more precisely $\widetilde{\mathcal{O}}(N_1^{2.5})$ as in Fukuchi et al. (2020)).

### 4.2 DETECTION

We now discuss ways the audit can detect manipulation of the sampling procedure. Recall that the audit has previously been given access to $f(D_0), f(D_1)$ representing the model outputs across all instances in the private dataset. The audit will then be given sub-samples $S_0', S_1'$ of $D_0, D_1$ on which they can compute the output of the model and compare with $f(D_0), f(D_1)$. To assess whether or not the sub-samples provided by the company were sampled uniformly at random, the audit has to conduct statistical tests. The null hypothesis of these tests will be that $S_0', S_1'$ were sampled uniformly at random from $D_0, D_1$. The detection Algorithm 2 with significance $\alpha$ uses both the Kolmogorov-Smirnov and Wald tests with Bonferonni corrections (*i.e.* the $\alpha/4$ terms in the Algorithm). The Kolmogorov-Smirnov and Wald tests are discussed in more detail in Appendix C.

---

**Algorithm 1** Compute non-uniform weights

---

1: **procedure** COMPUTE_WEIGHTS($D_1, \{\widehat{\Phi}_s(f, S_0', \boldsymbol{z}^{(j)})\}_j, \lambda$)
2:      $\beta$   $:= \text{sign}[\sum_{\boldsymbol{z}^{(j)} \in D_1} \widehat{\Phi}_s(f, S_0', \boldsymbol{z}^{(j)})]$
3:      $\mathcal{B}$   $:= \mathcal{C}(D_1, \mathbf{1}/N_1)$                        ▷ Unmanipulated background
4:      $\mathcal{B}_{\boldsymbol{\omega}}' := \mathcal{C}(D_1, \boldsymbol{\omega})$              ▷ Manipulated background as a function of $\boldsymbol{\omega}$
5:      $\boldsymbol{\omega} = \arg\min_{\boldsymbol{\omega}} \ \ \beta \sum_{\boldsymbol{z}^{(j)} \in D_1} \omega_j \widehat{\Phi}_s(f, S_0', \boldsymbol{z}^{(j)}) + \lambda \mathcal{W}(\mathcal{B}, \mathcal{B}_{\boldsymbol{\omega}}')$     ▷ Optimization Problem
6:      **return** $\boldsymbol{\omega}$;

---

**Algorithm 2** Detection with significance $\alpha$

---

1: **procedure** DETECT_FRAUD($f(D_0), f(D_1), f(S_0'), f(S_1'), \alpha, M$)
2:      **for** $i = 0, 1$ **do**
3:          $f(S_i) \sim \mathcal{C}(f(D_i), \mathbf{1}/N_i)^M$                  ▷ Subsample without cheating.
4:          p-value-KS = KS( $f(S_i), f(S_i')$ )          ▷ KS test comparing $f(S_i)$ and $f(S_i')$
5:          p-value-Wald = Wald( $f(S_i'), f(D_i)$ )               ▷ Wald test
6:          **if** p-value-KS $< \alpha/4$ **or** p-value-Wald $< \alpha/4$ **then**      ▷ Reject the null hypothesis
7:              **return** 1
8:      **return** 0;

---

## 4.3 WHOLE PROCEDURE

The procedure returning the subsets $S_0', S_1'$ is presented in Algorithm 3. It conducts a log-space search between $\lambda_{\min}$ and $\lambda_{\max}$ for the $\lambda$ hyper-parameter (line 6) in order to explore the possible attacks. For each value of $\lambda$, the attacker runs Algorithm 1 to obtain $\mathcal{B}_{\boldsymbol{\omega}}'$ (line 7), then repeatedly samples $S_1' \sim \mathcal{B}_{\boldsymbol{\omega}}'^M$ (line 10) and attempts to detect the fraud (line 11). The attacker will choose $\mathcal{B}_{\boldsymbol{\omega}}'$ that minimizes the magnitude of $\widehat{\Phi}_s$ while having a detection rate below some threshold $\tau$ (line 12). An example of search over $\lambda$ on a real-world dataset is presented in Figure 3.

---

**Algorithm 3** Fool SHAP

---

1: **procedure** FOOL_SHAP($f, D_0, D_1, M, \lambda_{\min}, \lambda_{\max}, \tau, \alpha$)
2:      $S_0' \sim \mathcal{C}(D_0, \mathbf{1}/N_0)^M$                        ▷ $S_0'$ is sampled without cheating
3:      Compute $\widehat{\Phi}_s(f, S_0', \boldsymbol{z}^{(j)}) \quad \forall \boldsymbol{z}^{(j)} \in D_1$                ▷ cf. Section B.3
4:      $\mathcal{B}^{\star} = \mathcal{C}(D_1, \mathbf{1}/N_1)$
5:      $\Phi_s^{\star} = 1/N_1 \sum_{\boldsymbol{z}^{(j)} \in D_1} \widehat{\Phi}_s(f, S_0', \boldsymbol{z}^{(j)})$            ▷ Initialize the solution
6:      **for** $\lambda = \lambda_{\max}, \dots, \lambda_{\min}$ **do**
7:          $\boldsymbol{\omega} =$ COMPUTE_WEIGHTS($D_1, \{\widehat{\Phi}_s(f, S_0', \boldsymbol{z}^{(j)})\}_j, \lambda$)
8:          Detection = 0
9:          **for** rep $= 1, \dots, 100$ **do**                  ▷ Detect the manipulation
10:              $S_1' \sim \mathcal{B}_{\boldsymbol{\omega}}'^M$
11:              Detection += DETECT_FRAUD($f(D_0), f(D_1), f(S_0'), f(S_1'), \alpha, M$)
12:          **if** $|\sum_{\boldsymbol{z}^{(j)} \in D_1} \omega_j \widehat{\Phi}_s(f, S_0', \boldsymbol{z}^{(j)})| < |\Phi_s^{\star}|$ **and** Detection $< 100\tau$ **then**
13:              $\mathcal{B}^{\star} = \mathcal{B}_{\boldsymbol{\omega}}'$
14:              $\Phi_s^{\star} = \sum_{\boldsymbol{z}^{(j)} \in D_1} \omega_j \widehat{\Phi}_s(f, S_0', \boldsymbol{z}^{(j)})$           ▷ Update the solution
15:      $S_1' \sim \mathcal{B}^{\star M}$               ▷ Cherry-pick by sampling from the non-uniform background
16:      **return** $S_0', S_1'$

---

One limitation of Fool SHAP is that it manipulates a single sensitive feature. In Appendix E.4, we present a possible extension of Algorithm 1 to handle multiple sensitive features and present preliminary results of its effectiveness. A second limitation is that it only applies to "interventional" Shapley values which break feature correlations. This choice was made because most methods in the SHAP library[2] are "interventional". Future work should port Fool SHAP to "observational" Shapley values that use conditional expectations to remove features (Frye et al., 2020).

---

[2] except the TreeExplainer when no background data is provided

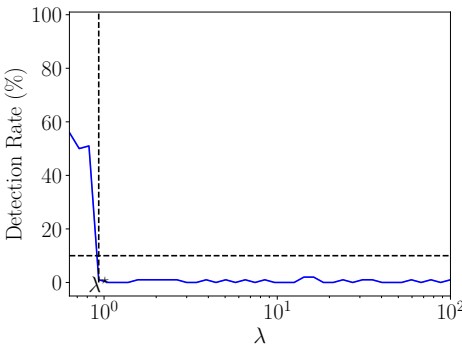 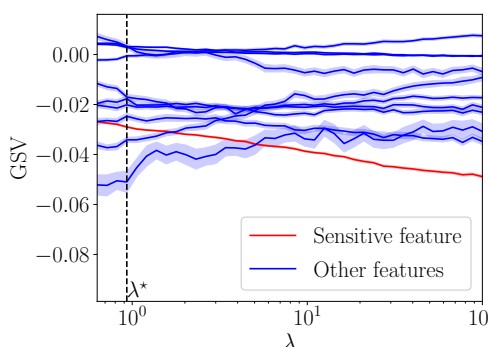

Figure 3: Example of log-space search over values of $\lambda$ using an XGBoost classifier fitted on Adults. (a) The detection rate as a function of the parameter $\lambda$ of the attack. The attacker uses a detection rate threshold $\tau = 10\%$. (b) For each value of $\lambda$, the vertical slice of the 11 curves is the GSV obtained with the resulting $\mathcal{B}'_{\boldsymbol{\omega}}$. The goal here is to reduce the amplitude of the sensitive feature (red curve).

## 4.4 CONTRIBUTIONS

The first technique to fool SHAP with perturbations of the background distribution was a genetic algorithm Baniecki & Biecek (2022). Although promising, the cross-over and mutation operations it employs to perturb data do not take into account feature correlations and can therefore generate unrealistic data. Moreover, the objective to minimize does not enforce similarity between the original and manipulated backgrounds. We show in Appendix E.3 that these limitations lead to systematic fraud detections. Hence, our contributions are two-fold. First, by perturbing the background via non-uniform weights over pre-existing instances (*i.e.* $\mathcal{B}'_{\boldsymbol{\omega}} := \mathcal{C}(D_1, \boldsymbol{\omega})$) rather than a genetic algorithm, we avoid the issue of non-realistic data. Second, by considering the Wasserstein distance, we can control the similarity between the original and fake backgrounds.

Since the Stealthity Biased Sampling technique introduced in Fukuchi et al. (2020) also leverages a non-uniform distribution over data points and the Wasserstein distance, it makes sense to adapt it to fool SHAP. Still, the approach of Fukuchi et al. is different from ours. Indeed, in their work, they minimize the Wasserstein distance while enforcing a hard constraint on the number of instances that land on the different bins for the target and sensitive feature, That way, they can set the Demographic Parity to any given value while staying close to the original data. In our setting of manipulating the model explanation, we leave the Demographic Parity intact and instead manipulate its feature attribution. In terms of the optimization objective, we now minimize a Shapley value with a soft constraint on the Wasserstein distance.

## 5 EXPERIMENTS

### 5.1 TOY EXPERIMENT

The task is predicting which individual will be hired for a job that requires carrying heavy objects. The causal graph for this toy data is presented in Figure 4 (left). We observe that sex ($S$) influences height ($H$), and that both these features influence the Muscular Mass ($M$). In the end, the hiring decisions ($Y$) are only based on the two attributes relevant to the job: $H$ and $M$. Also, two noise features $N1, N2$ were added. More details and justifications for this causal graph are discussed in Appendix D.1. Since strength and height (two important qualifications for applicants) are correlated with sex, any model $f$ that fits the data will exhibit some disparity in hiring rates between sexes. Although, if the model decisions do not rely strongly on feature $S$, the company can argue in favor of deployment. GSV are used by the audit to measure the amount by which the model relies on the sex feature, see Figure 4 (Middle). By employing Fool SHAP with $M = 100$, the company can reduce the GSV of feature $S$ considerably compared to the brute-force and genetic algorithms. More importantly, the audit is not able to detect that the provided samples $S'_0, S'_1$ were cherry-picked, see Figure 4 (Right). More results are presented in Appendix E.1.

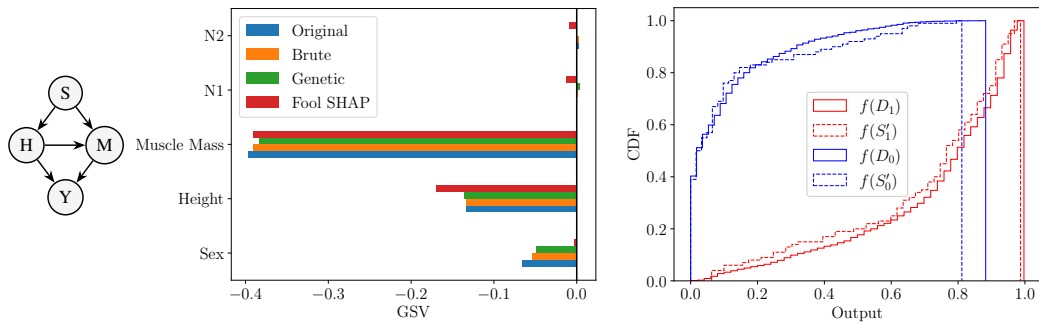

Figure 4: Toy example. Left: Causal graph. Middle: GSV for the different attacks with $M = 100$. Right: Comparison of the CDF of the Fool SHAP subsets $f(S'_0), f(S'_1)$ and the CDF over the whole data $f(D_0), f(D_1)$. Here the audit cannot detect the fraud using their detection algorithm.

## 5.2 DATASETS

We consider four standard datasets from the FAccT literature, namely COMPAS, Adult-Income, Marketing, and Communities.

- **COMPAS** regroups 6,150 records from criminal offenders in Florida collected from 2013-2014. This binary classification task consists in predicting who will re-offend within two years. The sensitive feature $s$ is `race` with values $x_s = 0$ for African-American and $x_s = 1$ for Caucasian.

- **Adult Income** contains demographic attributes of 48,842 individuals from the 1994 U.S. census. It is a binary classification problem with the goal of predicting whether or not a particular person makes more than 50K USD per year. The sensitive feature $s$ in this dataset is `gender`, which took values $x_s = 0$ for female, and $x_s = 1$ for male.

- **Marketing** involves information on 41,175 customers of a Portuguese bank and the binary classification task is to predict who will subscribe to a term deposit. The sensitive attribute is `age` and took values $x_s = 0$ for age `30-60`, and $x_s = 1$ for age `not30-60`

- **Communities & Crime** contains per-capita violent crimes for 1994 different communities in the US. The binary classification task is to predict which communities have crimes below the median rate. The sensitive attribute is `PercentWhite` and took values $x_s = 0$ for `PercentWhite<90%`, and $x_s = 1$ for `PercentWhite>=90%`.

Three models were considered for the two datasets: Multi-Layered Perceptrons (MLP), Random Forests (RF), and eXtreme Gradient Boosted trees (XGB). One model of each type was fitted on each dataset for 5 different train/test splits seeds, resulting in 60 models total. Values of the test set accuracy and demographic parity for each model type and dataset are presented in Appendix D.2.

## 5.3 DETECTOR CALIBRATION

Detector calibration refers to the assessment that, assuming the null hypothesis to be true, the probability of rejecting it (*i.e.* false positive) should be bounded by the significance level $\alpha$. Remember that the null hypothesis of the audit detector is that the sets $S'_0, S'_1$ provided by the company are sampled uniformly from $D_0, D_1$. Hence, to test the detector, the audit can sample their own subsets $f(S_0), f(S_1)$ uniformly from at random from $f(D_0), f(D_1)$, run the detection algorithm, and count the number of detection over 1000 repeats. Table 1 shows the false positive rates over the five train-test splits using a significance level $\alpha = 5\%$. We observe that the false positive rates are indeed bounded by $\alpha$ for all model types and datasets implying that the detector employed by the audit is calibrated.

Table 1: False Positive Rates (%) of the detector *i.e.* the frequency at which $S_0, S_1$ are considered cherry-picked when they are not. No rate should be above $5\%$.

|             | mlp | rf  | xgb |
|-------------|-----|-----|-----|
| COMPAS      | 4.0 | 4.6 | 4.0 |
| Adult       | 4.3 | 4.3 | 4.2 |
| Marketing   |     | 4.9 | 5.0 |
| Communities |     | 3.8 | 4.2 |

## 5.4 Attack Results and Discussion

The first step of the attack (line 3 of Algorithm 3) requires that the company run `SHAP` on their own and compute the necessary coefficients to run Algorithm 1. For the COMPAS and Adults datasets, the `ExactExplainer` of `SHAP` was used. Since Marketing and Communities contain more than 15 features, and since the `ExactExplainer` scales exponentially with the number of features, we were restricted to using the `TreeExplainer` (Lundberg et al., 2020) on these datasets. The `TreeExplainer` avoids the exponential cost of Shapley values but is only applicable to tree-based models such as RFs and XGBs. Therefore, we could not conduct the attack on MLPs fitted on Marketing and Communities.

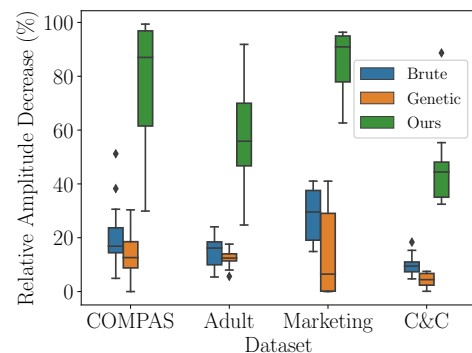

Figure 5: Relative decrease in amplitude of the sensitive feature attribution induced by the various attacks on `SHAP`.

The following step is to solve the MCF for various values of $\lambda$ (line 7 of Algorithm 3). As stated previously, solving the MCF can be done in polynomial time in terms of $N_1$, which was tractable for a small dataset like COMPAS and Communities, but not for larger datasets like Adult and Marketing. To solve this issue, as was done in Fukuchi et al. (2020), we compute the manipulated weights multiple times using 5 bootstrap sub-samples of $D_1$ of size 2000 to obtain a set of weights $\omega^{[1]}, \omega^{[2]}, \ldots, \omega^{[5]}$ which we average to obtain the final weights $\omega$.

Results of 46 attacks with $M = 200$ are shown in Figure 5. Specific examples of the conducted attacks are presented in Appendix E.2. As a point of reference, we also show results for the brute-force and genetic algorithms. To make comparisons to our attack more meaningful, the brute-force method was only allowed to run for the same amount of time it took to search for the non-uniform weights $\omega$ (about 30-180 seconds). Also, the genetic algorithm ran for 400 iterations and was stopped early if there were 10 consecutive detections. We note that, across all datasets, Fool SHAP leads to greater reductions of the sensitive feature attribution compared to brute-force search and the genetic perturbations of the background.

Now focusing on Fool SHAP, for the datasets COMPAS and Marketing, we observe median reductions in amplitudes of about $90\%$. This means that our attack can considerably reduce the apparent importance of the sensitive attribute. For the Adult and Communities datasets, the median reduction in amplitude is about $50\%$ meaning that we typically reduce by half the importance of the sensitive feature. Still, looking at the maximum reduction in amplitude for Adult-Income and Communities, we note that one attack managed to reduce the amplitude by $90\%$. Therefore, luck can play a part in the degree of success of Fool SHAP, which is to be expected from data-driven attacks.

Finally, the audit was consistently unable to detect the fraud using statistical tests. This observation raises concerns about the risk that `SHAP` explanations can be attacked to return not only manipulated attributions but also non-detectable fake evidence of fairness.

## 6 Conclusion

To conclude, we proposed a novel attack on Shapley values that does not require modifying the model but rather manipulates the sampling procedure that estimates expectations w.r.t the background distribution. We show on a toy example and four fairness datasets that our attack can reduce the importance of a sensitive feature when explaining the difference in outcomes between groups using `SHAP`. Crucially, the sampling manipulation is hard to detect by an audit that is given limited access to the data and model. These results raise concerns about the viability of using Shapley values to assess model fairness. We leave as future work the use of Shapley values to decompose other fairness metrics such as predictive equality and equal opportunity. Moreover, we wish to move to use cases beyond fairness, as we believe that the vulnerability of Shapley values that was demonstrated can apply to many other properties such as safety and security.

# 7 ETHICS STATEMENT

The main objective of this work is to raise awareness about the risk of manipulation of SHAP explanations and their undetectability. As such, it aims at exposing the potential negative societal impacts of relying on such explanations. It remains however possible that malicious model producers could use this attack to mislead end users or cheat during an audit. However, we believe this paper makes a significant step toward increasing the vigilance of the community and fostering the development of trustworthy explanations methods. Furthermore, by showing how fairness can be manipulated in explanation contexts, this work contributes to the research on the certification of the fairness of automated decision-making systems.

# 8 REPRODUCIBILITY STATEMENT

The source code of all our experiments is available online[3]. Moreover, experimental details are provided in appendix D.2 for the interested reader.

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

# A  PROOFS

## A.1  PROOFS FOR GLOBAL SHAPLEY VALUES (GSV)

**Proposition A.1** (**Proposition 2.1**). *The GSV have the following property*

$$\sum_{i=1}^{d} \Phi_i(f, \mathcal{F}, \mathcal{B}) = \underset{\boldsymbol{x} \sim \mathcal{F}}{\mathbb{E}}[f(\boldsymbol{x})] - \underset{\boldsymbol{x} \sim \mathcal{B}}{\mathbb{E}}[f(\boldsymbol{x})]. \tag{11}$$

*Proof.* As a reminder, we have defined the vector

$$\boldsymbol{\Phi}(f, \mathcal{F}, \mathcal{B}) = \underset{\substack{\boldsymbol{x} \sim \mathcal{F} \\ \boldsymbol{z} \sim \mathcal{B}}}{\mathbb{E}}[\boldsymbol{\phi}(f, \boldsymbol{x}, \boldsymbol{z})], \tag{12}$$

whose components sum up to

$$\sum_{i=1}^{d} \Phi_i(f, \mathcal{F}, \mathcal{B}) = \sum_{i=1}^{d} \underset{\substack{\boldsymbol{x} \sim \mathcal{F} \\ \boldsymbol{z} \sim \mathcal{B}}}{\mathbb{E}}[\phi_i(f, \boldsymbol{x}, \boldsymbol{z})] \tag{13}$$

$$= \underset{\substack{\boldsymbol{x} \sim \mathcal{F} \\ \boldsymbol{z} \sim \mathcal{B}}}{\mathbb{E}}\left[\sum_{i=1}^{d} \phi_i(f, \boldsymbol{x}, \boldsymbol{z})\right] \tag{14}$$

$$= \underset{\substack{\boldsymbol{x} \sim \mathcal{F} \\ \boldsymbol{z} \sim \mathcal{B}}}{\mathbb{E}}[f(\boldsymbol{x}) - f(\boldsymbol{z})] \tag{15}$$

$$= \underset{\boldsymbol{x} \sim \mathcal{F}}{\mathbb{E}}[f(\boldsymbol{x})] - \underset{\boldsymbol{z} \sim \mathcal{B}}{\mathbb{E}}[f(\boldsymbol{z})] \tag{16}$$

$$= \underset{\boldsymbol{x} \sim \mathcal{F}}{\mathbb{E}}[f(\boldsymbol{x})] - \underset{\boldsymbol{x} \sim \mathcal{B}}{\mathbb{E}}[f(\boldsymbol{x})], \tag{17}$$

where at the last step we have simply renamed a dummy variable. $\square$

**Proposition A.2** (**Proposition 4.1**). *Let $S_0'$ be **fixed**, and let $\xrightarrow{p}$ represent convergence in probability as the size $M$ of the set $S_1' \sim \mathcal{B}'^M$ increases, then we have*

$$\widehat{\Phi}_s(f, S_0', S_1') \xrightarrow{p} \sum_{j=1}^{N_1} \omega_j\, \widehat{\Phi}_s(f, S_0', \boldsymbol{z}^{(j)}). \tag{18}$$

*Proof.*

$$\widehat{\boldsymbol{\Phi}}(f, S_0', S_1') = \frac{1}{M^2} \sum_{\boldsymbol{x}^{(i)} \in S_0'} \sum_{\boldsymbol{z}^{(j)} \in S_1'} \boldsymbol{\phi}(f, \boldsymbol{x}^{(i)}, \boldsymbol{z}^{(j)})$$

$$= \frac{1}{M} \sum_{\boldsymbol{z}^{(j)} \in S_1'} \left(\frac{1}{M} \sum_{\boldsymbol{x}^{(i)} \in S_0'} \boldsymbol{\phi}(f, \boldsymbol{x}^{(i)}, \boldsymbol{z}^{(j)})\right) \tag{19}$$

$$= \frac{1}{M} \sum_{\boldsymbol{z}^{(j)} \in S_1'} \widehat{\boldsymbol{\Phi}}(f, S_0', \boldsymbol{z}^{(j)}).$$

Since $S_0'$ is assumed to be fixed, then the only random variable in $\widehat{\Phi}_s(f, S_0', \boldsymbol{z}^{(j)})$ is $\boldsymbol{z}^{(j)}$ which represents an instance sampled from the $\mathcal{B}'$. Therefore, we can define $\psi(\boldsymbol{z}) := \widehat{\Phi}_s(f, S_0', \boldsymbol{z})$ and we get

$$\widehat{\Phi}_s(f, S_0', S_1') = \frac{1}{M} \sum_{\boldsymbol{z}^{(j)} \in S_1'} \widehat{\Phi}_s(f, S_0', \boldsymbol{z}^{(j)})$$

$$= \frac{1}{M} \sum_{\boldsymbol{z}^{(j)} \in S_1'} \psi(\boldsymbol{z}^{(j)}) \qquad \text{with } S_1' \sim \mathcal{B}'^M. \tag{20}$$

By the weak law of large number, the following holds as $M$ goes to infinity (Wasserman, 2004, Theorem 5.6)

$$\frac{1}{M} \sum_{z^{(j)} \in S_1'} \psi(z^{(j)}) \xrightarrow{p} \mathbb{E}_{z \sim \mathcal{B}'}[\psi(z)]. \tag{21}$$

Now, as a reminder, the manipulated background distribution is $\mathcal{B}' := \mathcal{C}(D_1, \omega)$ with $\omega \neq \mathbf{1}/N_1$. Therefore

$$\begin{aligned}
\widehat{\Phi}_s(f, S_0', S_1') &\xrightarrow{p} \mathbb{E}_{z \sim \mathcal{B}'}[\psi(z)] \\
&= \mathbb{E}_{z \sim \mathcal{C}(D_1, \omega)}[\psi(z)] \\
&= \sum_{j=1}^{N_1} \omega_j \psi(z^{(j)}) \\
&= \sum_{j=1}^{N_1} \omega_j \widehat{\Phi}_s(f, S_0', z^{(j)})
\end{aligned} \tag{22}$$

concluding the proof. $\qquad\square$

### A.2 PROOFS FOR OPTIMIZATION PROBLEM

#### A.2.1 TECHNICAL LEMMAS

We provide some technical lemmas that will be essential when proving Theorem A.1. These are presented for completeness and are not intended as contributions by the authors. Let us first write the formal definition of the minimum of a function.

**Definition A.1** (Minimum). *Given some function $f : D \to \mathbb{R}$, the minimum of $f$ over $D$ (denoted $f^\star$) is defined as follows:*

$$f^\star = \min_{x \in D} f(x) \iff \exists x^\star \in D \text{ s.t. } f^\star = f(x^\star) \leqslant f(x) \quad \forall x \in D.$$

Basically, the notion of minimum coincides with the infimum $\inf f(D)$ (highest lower bound) when this lower bound is attained for some $x^\star \in D$. By the Extreme Values Theorem, the minimum always exists when $D$ is compact and $f$ is continuous. For the rest of this appendix, we shall only study optimization problems where points on the domain set $D = \{(x, y) : x \in \mathcal{X}, y \in \mathcal{Y}_x \subset \mathcal{Y}\}$ can be *selected* by the following procedure

1. Choose some $x \in \mathcal{X}$
2. Given the selected $x$, choose some $y \in \mathcal{Y}_x \subset \mathcal{Y}$, where the set $\mathcal{Y}_x$ is non-empty and depends on the value of $x$.

When optimizing functions over these domains, one can optimize in two steps as highlighted in the following lemma.

**Lemma A.1.** *Given a compact domain $D$ of the form described above and a continuous objective function $f : D \to \mathbb{R}$, the minimum $f^\star$ is attained for some $(x^\star, y^\star)$ and the following holds*

$$\min_{(x,y) \in D} f(x, y) = \min_{x \in \mathcal{X}} \min_{y \in \mathcal{Y}_x} f(x, y).$$

*Proof.* Let $\widetilde{f}(x) := \inf_{y \in \mathcal{Y}_x} f(x, y)$, which is a well defined function on $\mathcal{X}$. We can then take its infimum $f^\star = \inf_{x \in \mathcal{X}} \widetilde{f}(x)$. But is $f^\star$ an infimum of $f(D)$? By the definition of infimum

$$\begin{aligned} f^\star &\leqslant \widetilde{f}(x) && \forall \, x \in \mathcal{X} \\ &= \inf_{y \in \mathcal{Y}_x} f(x, y) \\ &\leqslant f(x, y) && \forall \, y \in \mathcal{Y}_x, \end{aligned}$$

so that $f^\star$ is a lower bound of $f(D)$. In fact, it is the highest lower bound possible so

$$\inf_{(x,y) \in D} f(x, y) = \inf_{x \in \mathcal{X}} \inf_{y \in \mathcal{Y}_x} f(x, y). \tag{23}$$

By the Extreme Value Theorem, since $D$ is compact and $f$ is continuous, there exists $(x^\star, y^\star) \in D$ s.t. $f^\star = \inf_{(x,y) \in D} f(x, y) = \max_{(x,y) \in D} f(x, y) = f(x^\star, y^\star)$. Since the infimum is attained on the left-hand-side of Equation 23, then it must also be attained on the right-hand-side and therefore we can replace all $\inf$ with $\min$ in Equation 23, leading to the desired result. $\square$

**Lemma A.2.** *Given a compact domain $D$ of the form described above and two continuous functions $h : \mathcal{X} \to \mathbb{R}$ and $g : \mathcal{Y} \to \mathbb{R}$, then*

$$\min_{(x,y) \in D} \left( h(x) + g(y) \right) = \min_{x \in \mathcal{X}} \left( h(x) + \min_{y \in \mathcal{Y}_x} g(y) \right)$$

*Proof.* Applying Lemma A.1 with the function $f(x, y) := h(x) + g(y)$ proves the Lemma. $\square$

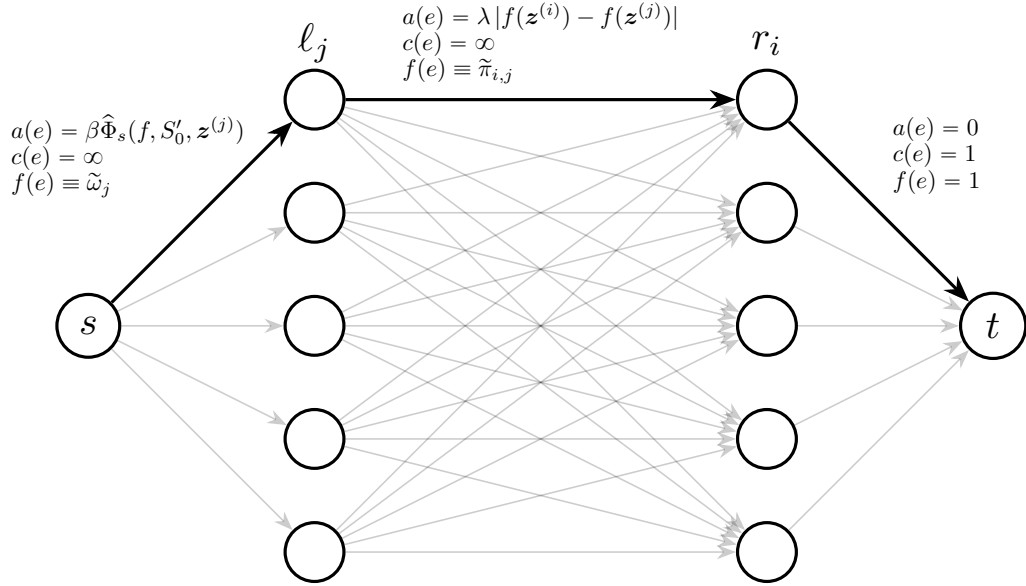

Figure 6: Graph $\mathbb{G}$ on which we solve the MCF. Note that the total amount of flow is $d = N_1$ and there are $N_1$ left and right nodes $\ell_j, r_i$.

### A.2.2 MINIMUM COST FLOWS

Let $\mathbb{G} = (\mathcal{V}, \mathcal{E})$ be a graph with vertices $v \in \mathcal{V}$ with directed edges $e \in \mathcal{E} \subset \mathcal{V} \times \mathcal{V}$, $c : \mathcal{E} \to \mathbb{R}^+$ be a capacity and $a : \mathcal{E} \to \mathbb{R}$ be a cost. Moreover, let $s, t \in \mathcal{E}$ be two special vertices called the source and the sink respectively, and $d \in \mathbb{R}^+$ be a total flow. The Minimum-Cost Flow (MCF) problem of $\mathbb{G}$ consists of finding the flow function $f : \mathcal{E} \to \mathbb{R}^+$ that minimizes the total cost

$$
\begin{aligned}
\min_f \quad & \sum_{e \in \mathcal{E}} a(e) f(e) \\
\text{s.t.} \quad & 0 \leqslant f(e) \leqslant c(e) \ \forall e \in \mathcal{E} \\
& \sum_{e \in u^+} f(e) - \sum_{e \in u^-} f(e) = \begin{cases} 0 & u \in \mathcal{V} \setminus \{s, t\} \\ d & u = s \\ -d & u = t \end{cases}
\end{aligned}
\tag{24}
$$

where $u^+ := \{(u, v) \in \mathcal{E}\}$ and $u^- := \{(v, u) \in \mathcal{E}\}$ are the outgoing and incoming edges from $u$. The terminology of *flow* arises from the constraint that, for vertices that are not the source nor the sink, the outgoing flow must equal the incoming one, which is reminiscent of conservation laws in fluidic. We shall refer to $f((u, v))$ as the flow from $u$ to $v$.

Now that we have introduced minimum cost flows, let us specify the graph that will be employed to manipulate GSV, see Figure 6. We label the flow going from the sink $s$ to one of the left vertices as $\widetilde{\omega}_i \equiv \omega_i \times N_1$, and the flow going from $\ell_j$ to $r_i$ as $\widetilde{\pi}_{i,j} \equiv \pi_{i,j} \times N_1$. The required flow is fixed at $d = N_1$.

**Theorem A.1.** *Solving the MCF of Figure 6 leads to a solution of the linear program in Algorithm 1.*

*Proof.* We begin by showing that the flow conservation constraints in the MCF imply that $\pi$ is a coupling measure (*i.e.* $\pi \in \Delta(\mathcal{B}, \mathcal{B}'_\omega)$), and $\omega$ is constrained to the probability simplex $\Delta(N_1)$. Applying the conservation law on the left-side of the graph leads to the conclusion that the flows entering vertices $\ell_j$ must sum up to $N_1$

$$\sum_{j=1}^{N_1} \widetilde{\omega}_j = N_1.$$

This implies that $\omega$ is must be part of the probability simplex. By conservation, the amount of flow that leaves a specific vertex $\ell_j$ must also be $\widetilde{\omega}_j$, hence

$$\sum_i \widetilde{\pi}_{ij} = \widetilde{\omega}_j.$$

For any edge outgoing from $r_i$ to the sink $t$, the flow must be exactly 1. This is because we have $N_1$ edges with capacity $c(e) = 1$ going into the sink and the sink must receive an incoming flow of $N_1$. As a consequence of the conservation law on a specific vertex $r_i$, the amount of flow that goes into each $r_i$ is also 1

$$\sum_j \widetilde{\pi}_{ij} = 1.$$

Putting everything together, from the conservation laws on $\mathbb{G}$, we have that $\omega \in \Delta(N_1)$, and $\pi \in \Delta(\mathcal{B}, \mathcal{B}'_\omega)$. Now, to make the parallel between the MCF and Algorithm 1, we must use Lemma A.2. Note that $\omega$ is restricted to the probability simplex, while $\pi$ is restricted to be a coupling measure. Importantly, the set of all possible coupling measures $\Delta(\mathcal{B}, \mathcal{B}'_\omega)$ is different for each $\omega$ because $\mathcal{B}'_\omega$ depends on $\omega$. Hence, the domain has the same structure as the ones tackled in Lemma A.2 (where $x \in \mathcal{X}$ becomes $\omega \in \Delta(N_1)$) and $y \in \mathcal{Y}_x$ becomes $\pi \in \Delta(\mathcal{B}, \mathcal{B}'_\omega)$). Also, the set of possible $\omega$ and $\pi$ is a bounded simplex in $\mathbb{R}^{N_1(N_1+1)}$ so it is compact, and the objective function of the MCF is linear, thus continuous. Hence, we can apply the Lemma A.2 to the MCF.

$$\min_f \sum_{e \in \mathcal{E}} f(e) a(e) = \min_{\widetilde{\omega}, \widetilde{\pi}} \sum_{j=1}^{N_1} \beta \widetilde{\omega}_j \widehat{\Phi}_s(f, S'_0, \boldsymbol{z}^{(j)}) + \lambda \sum_{i,j} \widetilde{\pi}_{ij} |f(\boldsymbol{z}^{(i)}) - f(\boldsymbol{z}^{(j)})|$$

$$= \min_{\widetilde{\omega}, \widetilde{\pi}} \frac{N_1}{N_1} \left( \beta \sum_{j=1}^{N_1} \widetilde{\omega}_j \widehat{\Phi}_s(f, S'_0, \boldsymbol{z}^{(j)}) + \lambda \sum_{i,j} \widetilde{\pi}_{ij} |f(\boldsymbol{z}^{(i)}) - f(\boldsymbol{z}^{(j)})| \right)$$

$$= N_1 \min_{\widetilde{\omega}, \widetilde{\pi}} \left( \beta \sum_{j=1}^{N_1} \frac{\widetilde{\omega}_j}{N_1} \widehat{\Phi}_s(f, S'_0, \boldsymbol{z}^{(j)}) + \lambda \sum_{i,j} \frac{\widetilde{\pi}_{ij}}{N_1} |f(\boldsymbol{z}^{(i)}) - f(\boldsymbol{z}^{(j)})| \right)$$

$$= N_1 \min_{\omega \in \Delta(N_1), \pi \in \Delta(\mathcal{B}, \mathcal{B}'_\omega)} \left( \beta \sum_{j=1}^{N_1} \omega_j \widehat{\Phi}_s(f, S'_0, \boldsymbol{z}^{(j)}) + \lambda \sum_{i,j} \pi_{i,j} |f(\boldsymbol{z}^{(i)}) - f(\boldsymbol{z}^{(j)})| \right)$$

$$= N_1 \min_{\omega \in \Delta(N_1), \pi \in \Delta(\mathcal{B}, \mathcal{B}'_\omega)} \Big( h(\omega) + g(\pi) \Big)$$

$$= N_1 \min_{\omega \in \Delta(N_1)} \left( h(\omega) + \min_{\pi \in \Delta(\mathcal{B}, \mathcal{B}')} g(\pi) \right) \qquad \text{(cf. Lemma A.2)}$$

$$= N_1 \min_{\omega \in \Delta(N_1)} \left( \beta \sum_{j=1}^{N_1} \omega_j \widehat{\Phi}_s(f, S'_0, \boldsymbol{z}^{(j)}) + \lambda \min_{\pi \in \Delta(\mathcal{B}, \mathcal{B}'_\omega)} \sum_{i,j} \pi_{i,j} |f(\boldsymbol{z}^{(i)}) - f(\boldsymbol{z}^{(j)})| \right)$$

$$= N_1 \min_{\omega \in \Delta(N_1)} \left( \beta \sum_{j=1}^{N_1} \omega_j \widehat{\Phi}_s(f, S'_0, \boldsymbol{z}^{(j)}) + \lambda \mathcal{W}(\mathcal{B}, \mathcal{B}'_\omega) \right)$$

which (up to a multiplicative constant $N_1$) is a solution of the linear program of Algorithm 1. □

# B  SHAPLEY VALUES

## B.1  LOCAL SHAPLEY VALUES (LSV)

We introduce Local Shapley Values (LSV) more formally. First, as explained earlier, Shapley values are based on coalitional game theory where the different features work together toward a common outcome $f(\boldsymbol{x})$. In a game, the features can either be present or absent, which is simulated by replacing some features with a baseline value $\boldsymbol{z}$.

**Definition B.1** (The Replace Function). *Let $\boldsymbol{x}$ be an input of interest $\boldsymbol{x}$, $S \subseteq \{1, 2, \ldots, d\}$ be a subset of input features that are considered active, and $\boldsymbol{z}$ be a baseline input, then the replace-function $\boldsymbol{r}_S : \mathbb{R}^d \times \mathbb{R}^d \to \mathbb{R}^d$ is defined as*

$$r_S(\boldsymbol{z}, \boldsymbol{x})_i = \begin{cases} x_i & \text{if } i \in S \\ z_i & \text{otherwise.} \end{cases} \tag{25}$$

*We note that this function is meant to "activate" the features in $S$.*

Now, if we let $\pi$ be a random permutation of $d$ features, and $\pi_i$ denote all features that appear before $i$ in $\pi$, the LSV are computed via

$$\phi_i(f, \boldsymbol{x}, \boldsymbol{z}) := \underset{\pi \sim \Omega}{\mathbb{E}} \big[ f(\boldsymbol{r}_{\pi_i \cup \{i\}}(\boldsymbol{z}, \boldsymbol{x})) - f(\boldsymbol{r}_{\pi_i}(\boldsymbol{z}, \boldsymbol{x})) \big], \quad i = 1, 2, \ldots, d, \tag{26}$$

where $\Omega$ is the uniform distribution over $d!$ permutations. Observe that the computation of LSV is scales poorly with the number of features $d$ hence model-agnostic computations are only possible with datasets with few features such as COMPAS and Adult-Income. For datasets with larger amounts of features the `TreeExplainer` algorithm (Lundberg et al., 2020) can be used to compute the LSV (cf. Equation 26) in polynomial time given that one is explaining a tree-based model.

## B.2  CONVERGENCE

As a reminder, we are interested in estimating the GSV $\boldsymbol{\Phi} \equiv \boldsymbol{\Phi}(f, \mathcal{F}, \mathcal{B})$ which requires estimating expectations w.r.t the foreground and background distributions. Said estimations can be conducted with Monte-Carlo where we sample $M$ instances

$$S_0 \sim \mathcal{F}^M \qquad S_1 \sim \mathcal{B}^M, \tag{27}$$

and compute the plug-in estimates

$$\begin{aligned} \widehat{\boldsymbol{\Phi}}(f, S_0, S_1) &:= \boldsymbol{\Phi}(f, \mathcal{C}(S_0, \boldsymbol{1}/M), \mathcal{C}(S_1, \boldsymbol{1}/M)) \\ &= \frac{1}{M^2} \sum_{\boldsymbol{x}^{(i)} \in S_0} \sum_{\boldsymbol{z}^{(j)} \in S_1} \boldsymbol{\phi}(f, \boldsymbol{x}^{(i)}, \boldsymbol{z}^{(j)}). \end{aligned} \tag{28}$$

We now show that, $\widehat{\boldsymbol{\Phi}}(f, S_0, S_1)$ is a consistent and asymptotically normal estimate of $\boldsymbol{\Phi}(f, \mathcal{F}, \mathcal{B})$

**Proposition B.1.** *Let $f : \mathcal{X} \to [0, 1]$ be a black box, $\mathcal{F}$ and $\mathcal{B}$ be distributions on $\mathcal{X}$, and $\widehat{\boldsymbol{\Phi}} \equiv \widehat{\boldsymbol{\Phi}}(f, S_0, S_1)$ be the plug-in estimate of $\boldsymbol{\Phi} \equiv \boldsymbol{\Phi}(f, \mathcal{F}, \mathcal{B})$, the following holds for any $\delta \in \,]0, 1[$ and $k = 1, 2 \ldots, d$*

$$\lim_{M \to \infty} \mathbb{P}\left( |\widehat{\Phi}_k - \Phi_k| \geqslant \frac{F^{-1}_{\mathcal{N}(0,1)}(1 - \delta/2)}{2\sqrt{M}} \sqrt{\sigma_{10}^2 + \sigma_{01}^2} \right) = \delta,$$

*where $F^{-1}_{\mathcal{N}(0,1)}$ is the inverse Cumulative Distribution Function (CDF) of the standard normal distribution, $\sigma_{10}^2 = \mathbb{V}_{\boldsymbol{x} \sim \mathcal{F}}[\,\mathbb{E}_{\boldsymbol{z} \sim \mathcal{B}}[\phi_i(f, \boldsymbol{x}, \boldsymbol{z})]\,]$ and $\sigma_{01}^2 = \mathbb{V}_{\boldsymbol{z} \sim \mathcal{B}}[\,\mathbb{E}_{\boldsymbol{x} \sim \mathcal{F}}[\phi_i(f, \boldsymbol{x}, \boldsymbol{z})]\,]$.*

*Proof.* The proof consists simply in noting that LSV $\phi_k(f, \boldsymbol{x}^{(i)}, \boldsymbol{z}^{(j)})$ are a function of two independent samples $\boldsymbol{x}^{(i)} \sim \mathcal{F}$ and $\boldsymbol{z}^{(j)} \sim \mathcal{B}$. The model $f$ is assumed fixed and hence for any feature $k$ we can define $h(\boldsymbol{x}^{(i)}, \boldsymbol{z}^{(j)}) := \phi_k(f, \boldsymbol{x}^{(i)}, \boldsymbol{z}^{(j)})$. Now, the estimates of GSV can be rewritten

$$\widehat{\Phi}_k(f, S_0, S_1) = \frac{1}{|S_0| |S_1|} \sum_{\boldsymbol{x}^{(i)} \in S_0} \sum_{\boldsymbol{z}^{(j)} \in S_1} h(\boldsymbol{x}^{(i)}, \boldsymbol{z}^{(j)}), \tag{29}$$

which we recognize as a well-known class of statistics called two-samples U-statistics. Such statistics are unbiased and asymptotically normal estimates of

$$\Phi_k(f, \mathcal{F}, \mathcal{B}) = \underset{\substack{\boldsymbol{x} \sim \mathcal{F} \\ \boldsymbol{z} \sim \mathcal{B}}}{\mathbb{E}}[h(\boldsymbol{x}, \boldsymbol{z})]. \tag{30}$$

The asymptotic normality of two-samples U-statistics is characterized by the following Theorem (Lee, 2019, Section 3.7.1).

**Theorem B.1.** *Let $\widehat{\Phi}_k \equiv \widehat{\Phi}_k(f, S_0, S_1)$ be a two-samples U-statistic with $|S_0| = N, |S_1| = M$, moreover let $h(\boldsymbol{x}, \boldsymbol{z})$ have finite first and second moments, then the following holds for any $\delta \in \,]0, 1[$*

$$\lim_{\substack{N+M \to \infty \\ s.t.\ N/(N+M) \to p \in (0,1)}} \mathbb{P}\left( |\widehat{\Phi}_k - \Phi_k| \geqslant \frac{F_{\mathcal{N}(0,1)}^{-1}(1 - \delta/2)}{\sqrt{M+N}} \sqrt{\frac{\sigma_{10}^2}{p} + \frac{\sigma_{01}^2}{1-p}} \right) = \delta,$$

*where $\sigma_{10}^2 = \mathbb{V}_{\boldsymbol{x} \sim \mathcal{F}}[\,\mathbb{E}_{\boldsymbol{z} \sim \mathcal{B}}[h(\boldsymbol{x}, \boldsymbol{z})]\,]$ and $\sigma_{01}^2 = \mathbb{V}_{\boldsymbol{z} \sim \mathcal{B}}[\,\mathbb{E}_{\boldsymbol{x} \sim \mathcal{F}}[h(\boldsymbol{x}, \boldsymbol{z})]\,]$.*

**Proposition B.1** follows from this Theorem by choosing $N = M, p = 0.5$ and noticing that having a model with bounded outputs ($f : \mathcal{X} \to [0, 1]$) implies that $|h(\boldsymbol{x}, \boldsymbol{z})| \leqslant 1 \;\forall \boldsymbol{x}, \boldsymbol{z} \in \mathcal{X}$ which means that $h(\boldsymbol{x}, \boldsymbol{z})$ has bounded first and second moments. □

### B.3 COMPUTE THE LSV

Running Algorithm 1 requires computing the coefficients $\widehat{\Phi}_s(f, S_0', \boldsymbol{z}^{(j)})$ for $j = 1, 2, \ldots, N_1$. To compute them, first note that they can be written in terms of LSV for all instances in $S_0'$

$$\widehat{\Phi}_s(f, S_0', \boldsymbol{z}^{(j)}) = \frac{1}{M} \sum_{\boldsymbol{x}^{(i)} \in S_0'} \phi_s(f, \boldsymbol{x}^{(i)}, \boldsymbol{z}^{(j)}). \tag{31}$$

The LSV $\phi_s(f, \boldsymbol{x}^{(i)}, \boldsymbol{z}^{(j)})$ are computed deeply in the SHAP code and are not directly accessible using the current API. Hence, we had to access them using Monkey-Patching *i.e.* we modified the `ExactExplainer` class so that it stores the LSV as one of its attributes. The attribute can then be accessed as seen in Figure 7. The code is provided as a fork the SHAP repository. For the `TreeExplainer`, because its source code is in C++ and wrapped in Python, we found it simpler to simply rewrite our own version of the algorithm in C++ so that it directly returns the LSV, instead of Monkey-Patching the `TreeExplainer`.

```
# Mask features using the whole background distribution
mask = Independent(D_1, max_samples=len(D_1))
explainer = shap.explainers.Exact(model.predict_proba, mask)
# Explain all instances sampled from the foreground
explainer(S_0)
# The LSV are extracted with Monkey-Patching
LSV = explainer.LSV          # LSV.shape = (n_features, |S_0|, |D_1|)
Phi_S_0_zj = LSV.mean(1).T  # Phi_S_0_zj.shape = (|D_1|, n_features)
```

Figure 7: How we extract the LSV from the `ExactExplainer` via Monkey-Patching.

# C  STATISTICAL TESTS

## C.1  KS TEST

A first test that can be conducted is a two-samples Kolmogorov-Smirnov (KS) test (Massey Jr, 1951). If we let

$$\widehat{F}_S(x) = \frac{1}{|S|} \sum_{z \in S} \mathbb{1}(z \leqslant x) \tag{32}$$

be the empirical CDF of observations in the set $S$. Given two sets $S$ and $S'$, the KS statistic is

$$\text{KS}(S, S') = \sup_{x \in \mathbb{R}} |\widehat{F}_S(x) - \widehat{F}_{S'}(x)|. \tag{33}$$

Under the null-hypothesis $H_0 : S \sim \mathcal{D}^{|S|}, S' \sim \mathcal{D}^{|S'|}$ for some univariate distribution $\mathcal{D}$, this statistic is expected to not be too large with high probability. Hence, when the company provides the subsets $S_0', S_1'$, the audit can sample their own two subsets $f(S_0), f(S_1)$ uniformly at random from $f(D_0), f(D_1)$ and compute the statistics $\text{KS}(f(S_0), f(S_0'))$ and $\text{KS}(f(S_1), f(S_1'))$ to detect a fraud.

## C.2  WALD TEST

An alternative is the Wald test, which is based on the central limit theorem. If $S_1 \sim \mathcal{B}^M$, then the empirical average of the model output over $S_1$ is asymptotically normally distributed as $M$ increases

$$\text{Wald}(f(S_1), f(\mathcal{B})) := \frac{\frac{1}{M} \sum_{z \in f(S_1)} z - \mu}{\sigma/\sqrt{M}} \rightsquigarrow \mathcal{N}(0, 1), \tag{34}$$

where $\mu := \mathbb{E}_{z \sim f(\mathcal{B})}[z]$ and $\sigma^2 := \mathbb{V}_{z \sim f(\mathcal{B})}[z]$ are the expected value and variance of the model output across the whole background. The same reasoning holds for $S_0$ and the foreground $\mathcal{F}$. Applying the Wald test with significance $\alpha$ would detect fraud when

$$|\text{Wald}(f(S_1'), f(\mathcal{B}))| > F_{\mathcal{N}(0,1)}^{-1}(1 - \alpha/2), \tag{35}$$

where $F_{\mathcal{N}(0,1)}^{-1}$ is the inverse of the CDF of a standard normal variable.

# D   METHODOLOGICAL DETAILS

## D.1   TOY EXAMPLE

The toy dataset was constructed to closely match the results of the following empirical study comparing skeletal mass distributions between men and women (Janssen et al., 2000). Firstly, the sex feature was sampled from a Bernoulli

$$S \sim \text{Bernoulli}(0.5). \tag{36}$$

According to Table 1 of Janssen et al. (2000), the average height of women participants was 163 cm while it was 177cm for men. Both height distributions had the same standard deviation of 7cm. Hence we sampled height via

$$\begin{aligned} H|S=\texttt{man} &\sim \mathcal{N}(177, 49) \\ H|S=\texttt{woman} &\sim \mathcal{N}(163, 49) \end{aligned} \tag{37}$$

It was noted in Janssen et al. (2000) that there was approximately a linear relationship between height and skeletal muscle mass for both sexes. Therefore, we computed the muscle mass $M$ as

$$\begin{aligned} M|\{H=h, S=\texttt{man}\} &= 0.186h + 5\epsilon \\ M|\{H=h, S=\texttt{woman}\} &= 0.128h + 4\epsilon \\ \text{with } \epsilon &\sim \mathcal{N}(0, 1) \end{aligned} \tag{38}$$

The values of coefficients 0.186, 0.128 and noise levels 5 and 4 were chosen so the distributions of $M|S$ would approximately match that of Table 1 in Janssen et al. (2000). Finally the target was chosen following

$$\begin{aligned} Y|\{H=h, M=m\} &\sim \text{Bernoulli}(\, P(H, M)\, ) \\ \text{with } P(H, M) &= \left[ 1 + \exp\{100 \times \mathbb{1}(H < 160) - 0.3(M - 28)\} \right]^{-1}. \end{aligned} \tag{39}$$

Simply put, the chances of being hired in the past ($Y$) were impossible for individuals with a smaller height than 160cm. Moreover, individuals with a higher mass skeletal mass were given more chances to be admitted. Yet, individuals with less muscle mass could still be given the job if they displayed sufficient determination. In the end, we generated 6000 samples leading to the following disparity in $Y$.

$$\mathbb{P}(Y = 1|S=\texttt{man}) = 0.733 \qquad \mathbb{P}(Y = 1|S=\texttt{woman}) = 0.110. \tag{40}$$

Table 2: Models Test Accuracy % (mean $\pm$ stddev).

|  | mlp | rf | xgb |
|---|---|---|---|
| COMPAS | $68.2 \pm 0.9$ | $67.7 \pm 0.8$ | $68.6 \pm 0.8$ |
| Adult | $85.6 \pm 0.3$ | $86.3 \pm 0.2$ | $87.1 \pm 0.1$ |
| Marketing |  | $91.1 \pm 0.1$ | $91.4 \pm 0.3$ |
| Communities |  | $83 \pm 2$ | $82 \pm 2$ |

Table 3: Models Demographic Parity (mean $\pm$ stddev).

|  | mlp | rf | xgb |
|---|---|---|---|
| COMPAS | $-0.12 \pm 0.01$ | $-0.11 \pm 0.01$ | $-0.11 \pm 0.02$ |
| Adult | $-0.20 \pm 0.01$ | $-0.19 \pm 0.01$ | $-0.192 \pm 0.004$ |
| Marketing |  | $-0.104 \pm 0.005$ | $-0.11 \pm 0.01$ |
| Communities |  | $-0.50 \pm 0.01$ | $-0.54 \pm 0.02$ |

## D.2 REAL DATA

The datasets were first divided into train/test subsets with ratio $\frac{4}{5} : \frac{1}{5}$. The models were trained on the training set and evaluated on the test set. All categorical features for COMPAS, Adult, and Marketing were one-hot-encoded which resulted in a total of 11, 40, and 61 columns for each dataset respectively. A simple 50 steps random search was conducted to fine-tune the hyper-parameters with cross-validation on the training set. The resulting test set performance and demographic parities for all models and datasets, aggregated over 5 random data splits, are reported in Tables 2 and 3 respectively.

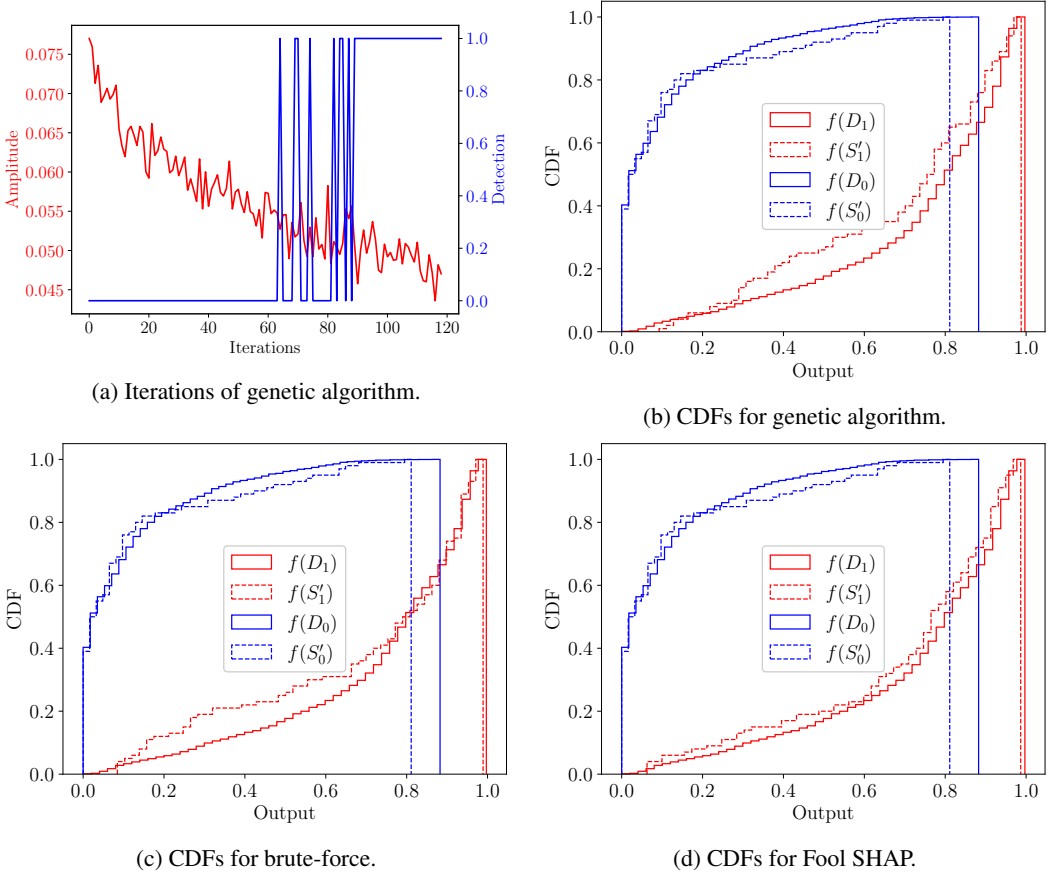

(a) Iterations of genetic algorithm.

(b) CDFs for genetic algorithm.

(c) CDFs for brute-force.

(d) CDFs for Fool SHAP.

Figure 8

# E    ADDITIONAL RESULTS

## E.1    TOY EXAMPLE

Figure 8 presents additional results for the toy example. More specifically, Figure 8 (a) illustrates the evolution of the detection and amplitude of the sensitive feature during the genetic algorithm. We note that beyond 90 iterations, the detector is systematically able to assert that the dataset is manipulated. The smallest value of amplitude that can be reached via the genetic algorithm without being detected is around 0.05. Figures 8 (b) (c) and (d) show the CDFs of $f(S_1')$ where $S_1'$ is chosen via the genetic algorithm, brute-force, and Fool SHAP respectively. We observe that Fool SHAP is the method where the resulting CDF for $f(S_1')$ is closest to the CDF for $f(D_1)$. This is why the audit is not able to detect fraud using statistical tests. The fact that Fool SHAP generates fake CDFs that are close to the data CDFs is a consequence of minimizing the Wasserstein distance. These results highlight the superiority of Fool SHAP compared to the brute-force approach and the genetic algorithm.

## E.2 EXAMPLES OF ATTACKS

In this section, we present 8 specific examples of the attacks that were conducted on COMPAS, Adult, Marketing, and Communities.

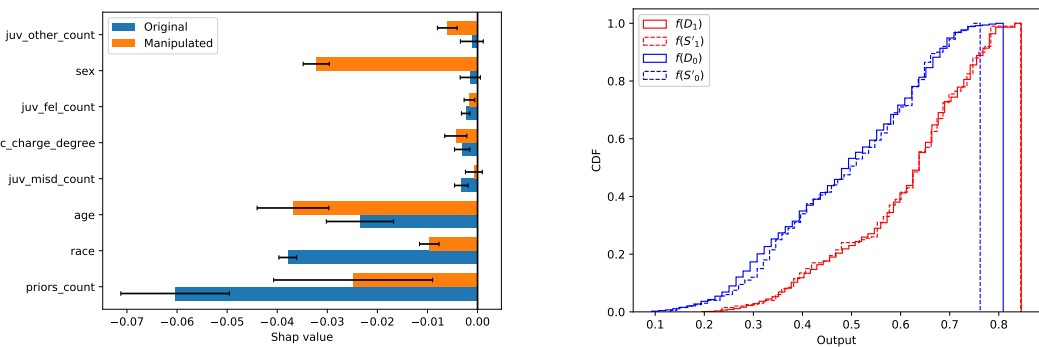

Figure 9: Attack of RF fitted on COMPAS. Left: GSV before and after the attack with $M = 200$. As a reminder, the sensitive attribute is `race`. Right: Comparison of the CDF of the misleading subsets $f(S'_0), f(S'_1)$ and the CDF over the whole data. $f(D_0), f(D_1)$.

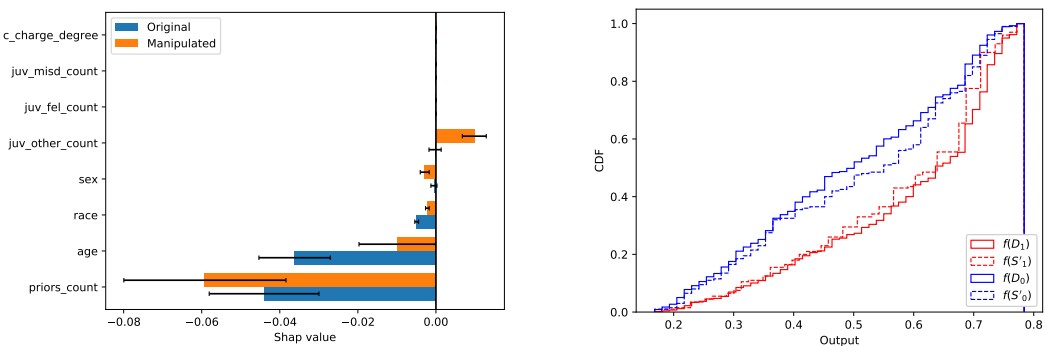

Figure 10: Attack of XGB fitted on COMPAS. Left: GSV before and after the attack with $M = 200$. As a reminder, the sensitive attribute is `race`. Right: Comparison of the CDF of the misleading subsets $f(S'_0), f(S'_1)$ and the CDF over the whole data. $f(D_0), f(D_1)$.

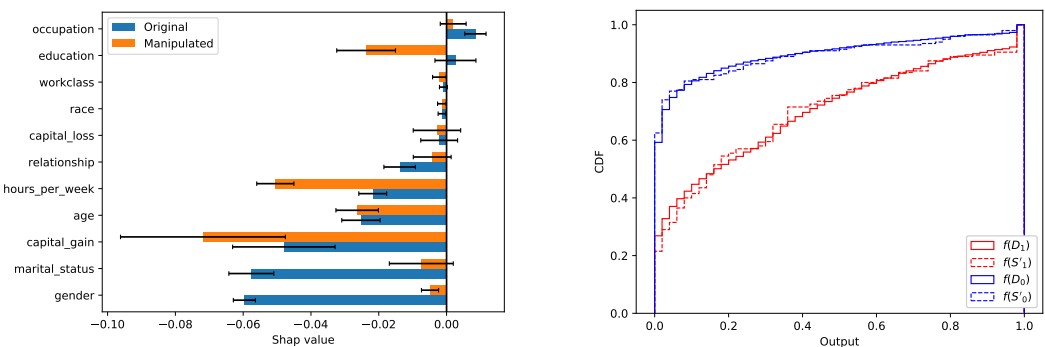

Figure 11: Attack of XGB fitted on Adults. Left: GSV before and after the attack with $M = 200$. As a reminder, the sensitive attribute is `gender`. Right: Comparison of the CDF of the misleading subsets $f(S'_0), f(S'_1)$ and the CDF over the whole data. $f(D_0), f(D_1)$.

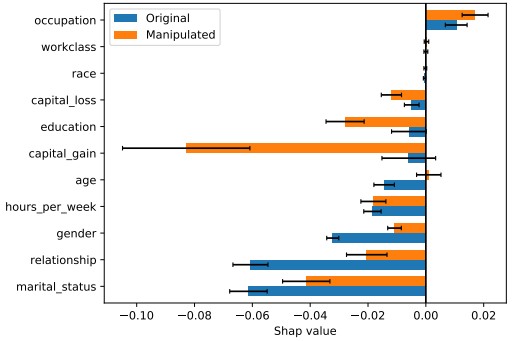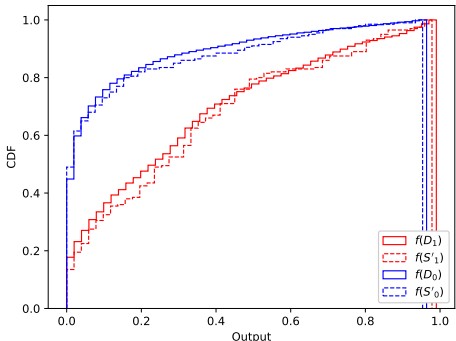

Figure 12: Attack of RF fitted on Adults. Left: GSV before and after the attack with $M = 200$. As a reminder, the sensitive attribute is gender. Right: Comparison of the CDF of the misleading subsets $f(S'_0), f(S'_1)$ and the CDF over the whole data. $f(D_0), f(D_1)$.

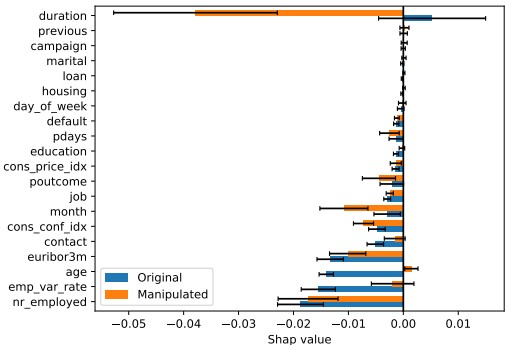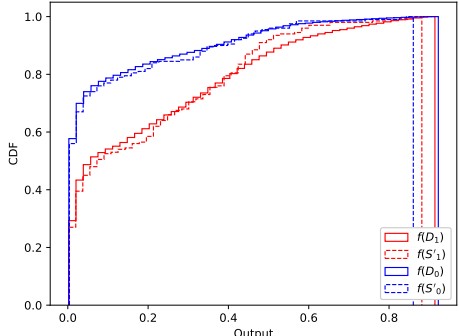

Figure 13: Attack of RF fitted on Marketing. Left: GSV before and after the attack with $M = 200$. As a reminder, the sensitive attribute is age. Right: Comparison of the CDF of the misleading subsets $f(S'_0), f(S'_1)$ and the CDF over the whole data. $f(D_0), f(D_1)$.

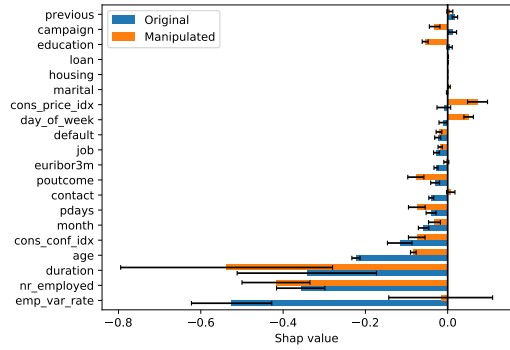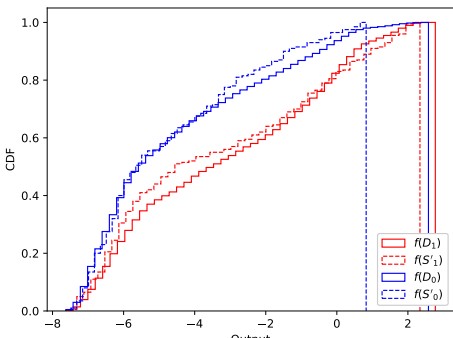

Figure 14: Attack of XGB fitted on Marketing. Left: GSV before and after the attack with $M = 200$. As a reminder, the sensitive attribute is age. Right: Comparison of the CDF of the misleading subsets $f(S'_0), f(S'_1)$ and the CDF over the whole data. $f(D_0), f(D_1)$.

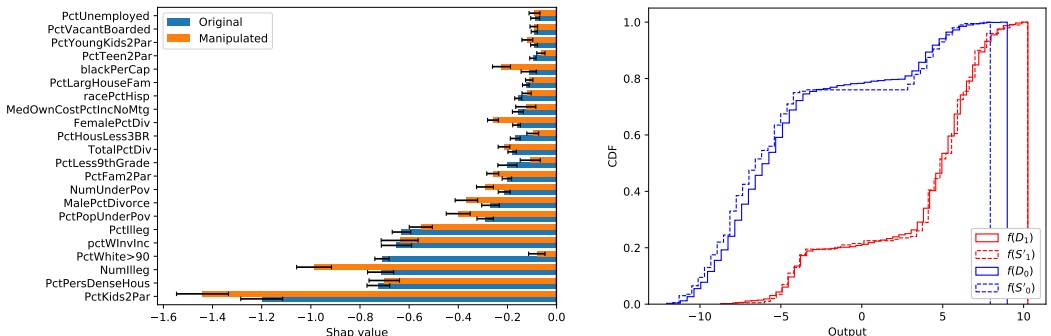

Figure 15: Attack of XGB fitted on Communities. Left: GSV before and after the attack with $M = 200$. As a reminder, the sensitive attribute is `PctWhite>90`. Right: Comparison of the CDF of the misleading subsets $f(S'_0), f(S'_1)$ and the CDF over the whole data. $f(D_0), f(D_1)$.

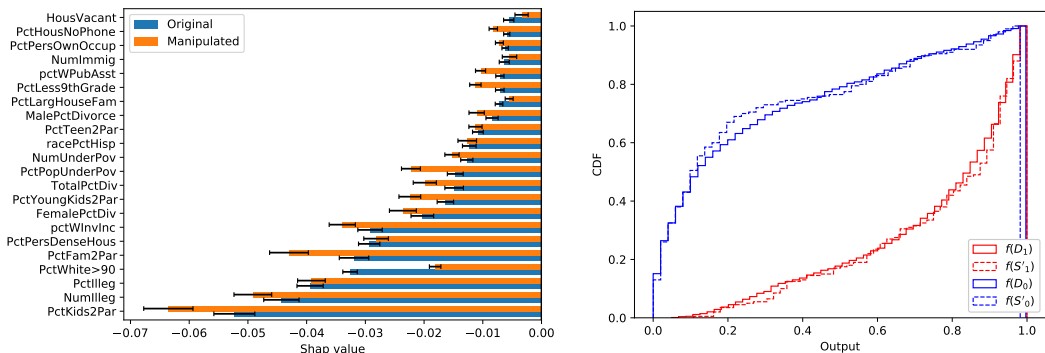

Figure 16: Attack of RF fitted on Communities. Left: GSV before and after the attack with $M = 200$. As a reminder, the sensitive attribute is `PctWhite>90`. Right: Comparison of the CDF of the misleading subsets $f(S'_0), f(S'_1)$ and the CDF over the whole data. $f(D_0), f(D_1)$.

### E.3 GENETIC ALGORITHM

This section motivates the use of stealthily biased sampling to perturb Shapley Values in place of the method of Baniecki et al. (2021), which fools `SHAP` by perturbing the background dataset $S'_1$ via a genetic algorithm. In said genetic algorithm, a population of $P$ fake background datasets $\{S'^{(k)}_1\}^P_{k=1}$ evolves iteratively following three biological mechanisms

- **Cross-Over:** Two parents produce two children by switching some of their feature values.

- **Mutation:** Some individuals are perturbed with small Gaussian noise.

- **Selection:** The individuals $S'^{(k)}_1$ with the smallest amplitudes $|\Phi_s(f, S'_0, S'^{(k)}_1)|$ are selected for the next generation.

Although the use of a genetic algorithm makes the method of Baniecki et al. (2021) very versatile, its main drawback is there is no constraint on the similarity between the perturbed background and the original one. Moreover, the mutation and cross-over operations ignore the correlations between features and hence the perturbed dataset can contain unrealistic instances. To highlight these issues, Figure 17 presents the first two principal components of $D_1$ and $S'_1$ for the XGB models used in Section 5.4. On COMPAS and Marketing especially, we see that the fake samples $S'_1$ lie in regions outside of the data manifold. For Adult-Income and Marketing, the fake data overlaps more with the original one, but this could be an artifact of only visualizing 2 dimensions.

For a more rigorous analysis of "similarity" between $S'_1$ and $D_1$, we must study the detection rate of the audit detector. To this end, Figures 18 and 19, present the amplitude reduction and the detection rate after a given number of iterations of the genetic algorithm. These curves show the average and

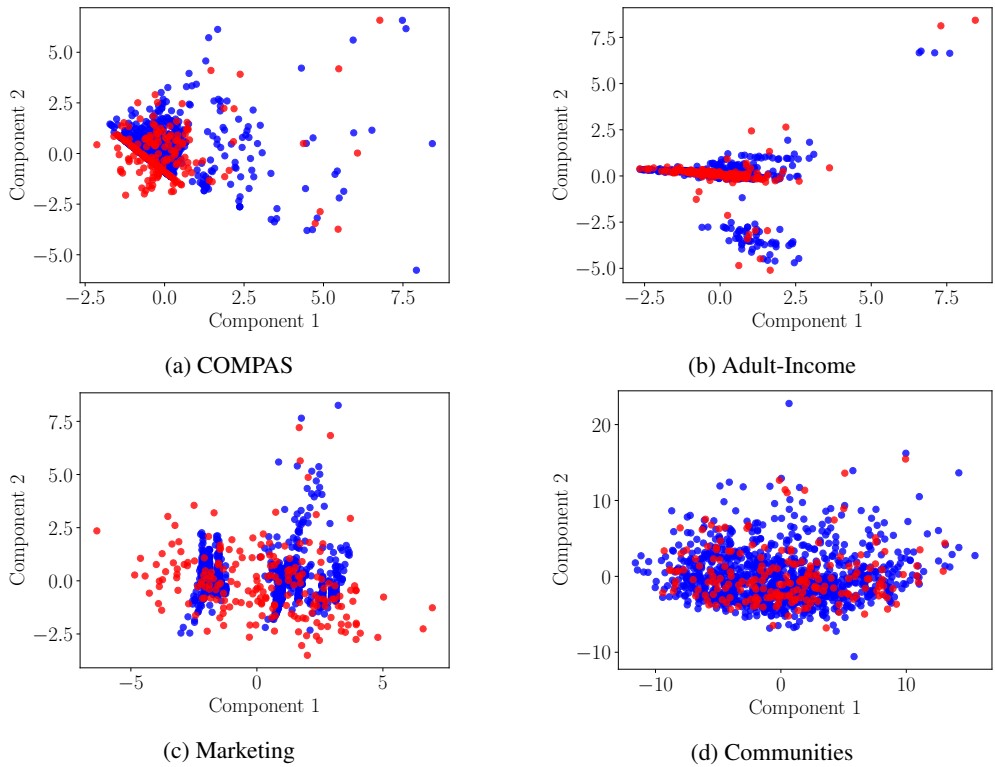

(a) COMPAS  (b) Adult-Income

(c) Marketing  (d) Communities

Figure 17: First two principal components of $D_1$ (Blue) and $S'_1$ (Red) returned by the genetic algorithm on XGB models.

standard deviation across the 5 train/test splits employed in our main experiments. Moreover, window 20 convolutions were used to smooth the curves and make them more readable. On the Marketing and Communities datasets, we see that for both XGB and Random Forests models, the detector is quickly able to assert that the data was manipulated. We suspect the genetic algorithm cannot fool the detector on these two datasets because they contain a large number of features (Marketing has 20, Communities has 98). Such a large number of features could make it harder to perturbate samples while staying close to the original data manifold. Since the model behavior is unpredictable outside of the data manifold, it is impossible for the genetic algorithm to guarantee that the CDF of $f(S'_1)$ will be close to the CDF of $f(D_1)$. For adult-income, the detection rate appears to be lower but still, the largest reductions in amplitude of the sensitive feature were about $15\%$, even after 2.5 hours of run-time.

Contrary to the genetic algorithm, our method Fool SHAP addresses both constraints of making the fake data realistic and keeping it close to the original dataset. Indeed, our objective is tuned to make sure that the Wasserstein distance between the original and perturbed data is small. Moreover, since we do not generate new samples but rather apply non-uniform weights to pre-existing ones, we do not run into the risk of generating unrealistic data.

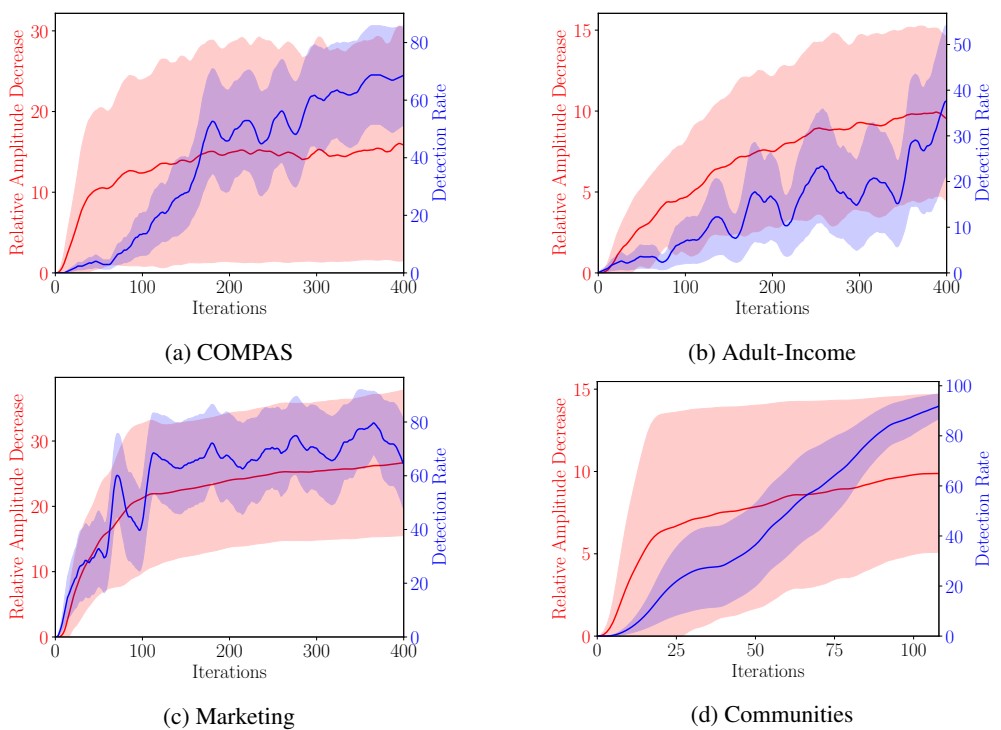

Figure 18: Iterations of the genetic algorithm applied to 5 XGB models per dataset.

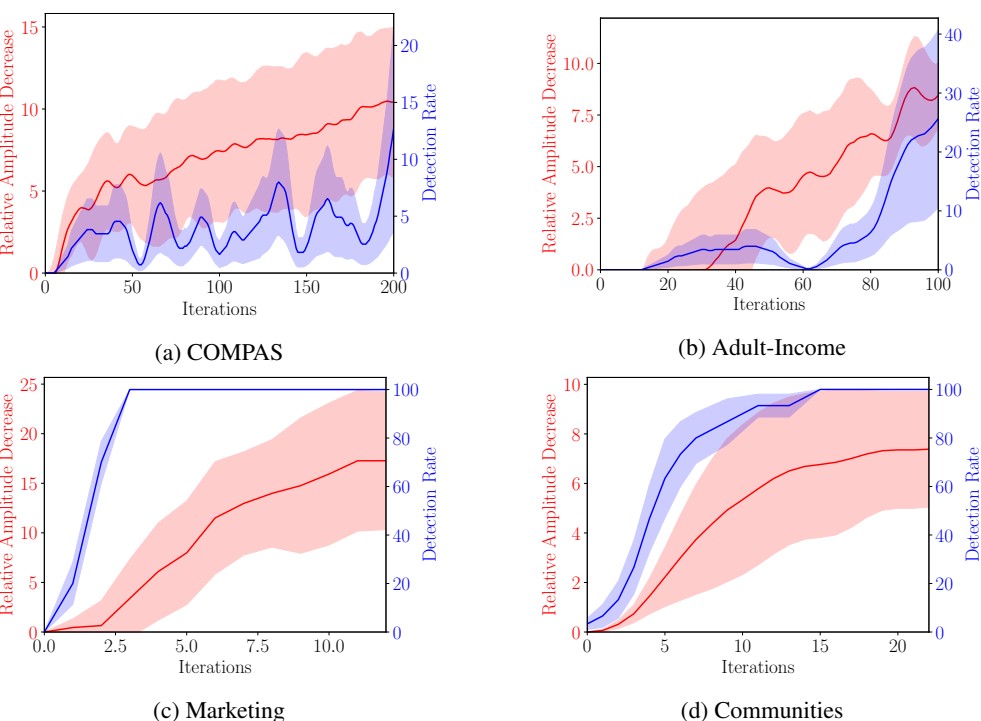

Figure 19: Iterations of the genetic algorithm applied to 5 RF models per dataset.

### E.4 MULTIPLE SENSITIVE ATTRIBUTES

We present preliminary results for settings where one wishes to manipulate the Shapley values of multiple sensitive features $s$ each part of a set $s \in \mathcal{S}$. For example, in our experiments, we considered `gender` as a sensitive attribute for the Adult-Income dataset and we showed that one can diminish the attribution of this feature. Nonetheless, there are two other features in Adult-Income that share information with `gender`: `relationship` and `marital-status`. Indeed, `relationship` can take the value `widowed` and `marital-status` can take the value `wife`, which are both proxies of `gender=female`. For this reason, these two other features may be considered sensitive and decision-making that relies strongly on them may not be acceptable. Hence, we must derive a method that reduces the total attributions of the features in $\mathcal{S} = \{\texttt{gender}, \texttt{relationship}, \texttt{marital-status}\}$.

We first let $\beta_s := \text{sign}\big[\sum_{\boldsymbol{z}^{(j)} \in D_1} \widehat{\Phi}_s(f, S_0', \boldsymbol{z}^{(j)})\big]$ for any $s \in \mathcal{S}$. In our experiments, all these signs will typically be negative. The proposed approach is to minimize the $\ell_1$ norm

$$\|(\widehat{\Phi}_s(f, S_0', S_1'))_{s \in \mathcal{S}}\|_1 := \sum_{s \in \mathcal{S}} |\widehat{\Phi}_s(f, S_0', S_1')|, \tag{41}$$

which we interpret as the total amount of disparity we can attribute to the sensitive attributes. Remember that $\widehat{\Phi}_s(f, S_0', S_1')$ converges in probability to $\sum_{\boldsymbol{z}^{(j)} \in D_1} \omega_j \widehat{\Phi}_s(f, S_0', \boldsymbol{z}^{(j)})$ (cf. Proposition 4.1). Therefore minimizing the $\ell_1$ norm will require minimizing

$$\sum_{s \in \mathcal{S}} \beta_s \sum_{\boldsymbol{z}^{(j)} \in D_1} \omega_j \widehat{\Phi}_s(f, S_0', \boldsymbol{z}^{(j)}) = \sum_{\boldsymbol{z}^{(j)} \in D_1} \omega_j \sum_{s \in \mathcal{S}} \beta_s \widehat{\Phi}_s(f, S_0', \boldsymbol{z}^{(j)}), \tag{42}$$

which is again a linear function of the weights. We present Algorithm 4 as an overload of Algorithm 1 that now supports taking multiple sensitive attributes as inputs.

---

**Algorithm 4** Compute non-uniform weights for multiple sensitive attributes $s \in \mathcal{S}$

1: **procedure** COMPUTE_WEIGHTS($D_1, \big\{\widehat{\Phi}_s(f, S_0', \boldsymbol{z}^{(j)})\big\}_{s,j}, \lambda$)
2: $\quad \beta_s := \text{sign}\big[\sum_{\boldsymbol{z}^{(j)} \in D_1} \widehat{\Phi}_s(f, S_0', \boldsymbol{z}^{(j)})\big] \quad \forall s \in \mathcal{S};$
3: $\quad \mathcal{B} := \mathcal{C}(D_1, \mathbf{1}/N_1)$                 $\triangleright$ Unmanipulated background
4: $\quad \mathcal{B}_{\boldsymbol{\omega}}' := \mathcal{C}(D_1, \boldsymbol{\omega})$          $\triangleright$ Manipulated background as a function of $\boldsymbol{\omega}$
5: $\quad \boldsymbol{\omega} = \arg\min_{\boldsymbol{\omega}} \ \sum_{\boldsymbol{z}^{(j)} \in D_1} \omega_j \sum_{s \in \mathcal{S}} \beta_s \widehat{\Phi}_s(f, S_0', \boldsymbol{z}^{(j)}) + \lambda \mathcal{W}(\mathcal{B}, \mathcal{B}_{\boldsymbol{\omega}}')$
6: $\quad$ **return** $\boldsymbol{\omega}$;

---

The only difference in the resulting MCF is that we must use the cost $a(e) = \sum_{s \in \mathcal{S}} \beta_s \widehat{\Phi}_s(f, S_0', \boldsymbol{z}^{(j)})$ for edges $(s, \ell_j)$ in the graph $\mathbb{G}$ of Figure 6. This new algorithm is guaranteed to diminish the $\ell_1$ norm of the attributions of all sensitive features. However, that this does not imply that all sensitive attributes will diminish in amplitude. Indeed, minimizing the sum of multiple quantities does not guarantee that each quantity will diminish. For example, $4 + 7$ is smaller than $6 + 6$ although $4$ is smaller than $6$ and $7$ is higher than $6$. Still, we see reducing the $\ell_1$ norm as a natural way to hide the total amount of disparity that is attributable to the sensitive features. Another important methodological change is the way we select the optimal hyper-parameter $\lambda$ in Algorithm 3. Now at line 12, we use the $\ell_1$ norm $\sum_{s \in \mathcal{S}} |\sum_{\boldsymbol{z}^{(j)} \in D_1} \omega_j \widehat{\Phi}_s(f, S_0', \boldsymbol{z}^{(j)})|$ as a selection criterion.

Figures 20 and 21 present preliminary results of attacks on three RFs/XGBs fitted on Adults with different train/test splits. We note that in all cases, before the attack, the three sensitive features had large negative attributions. By applying our method, we can considerably reduce the amplitude of the two sensitive attributes. The attribution of the remaining sensitive feature remains approximately constant or slightly becomes more negative. We leave it as future work to run large-scale experiments with multiple sensitive features for various datasets.

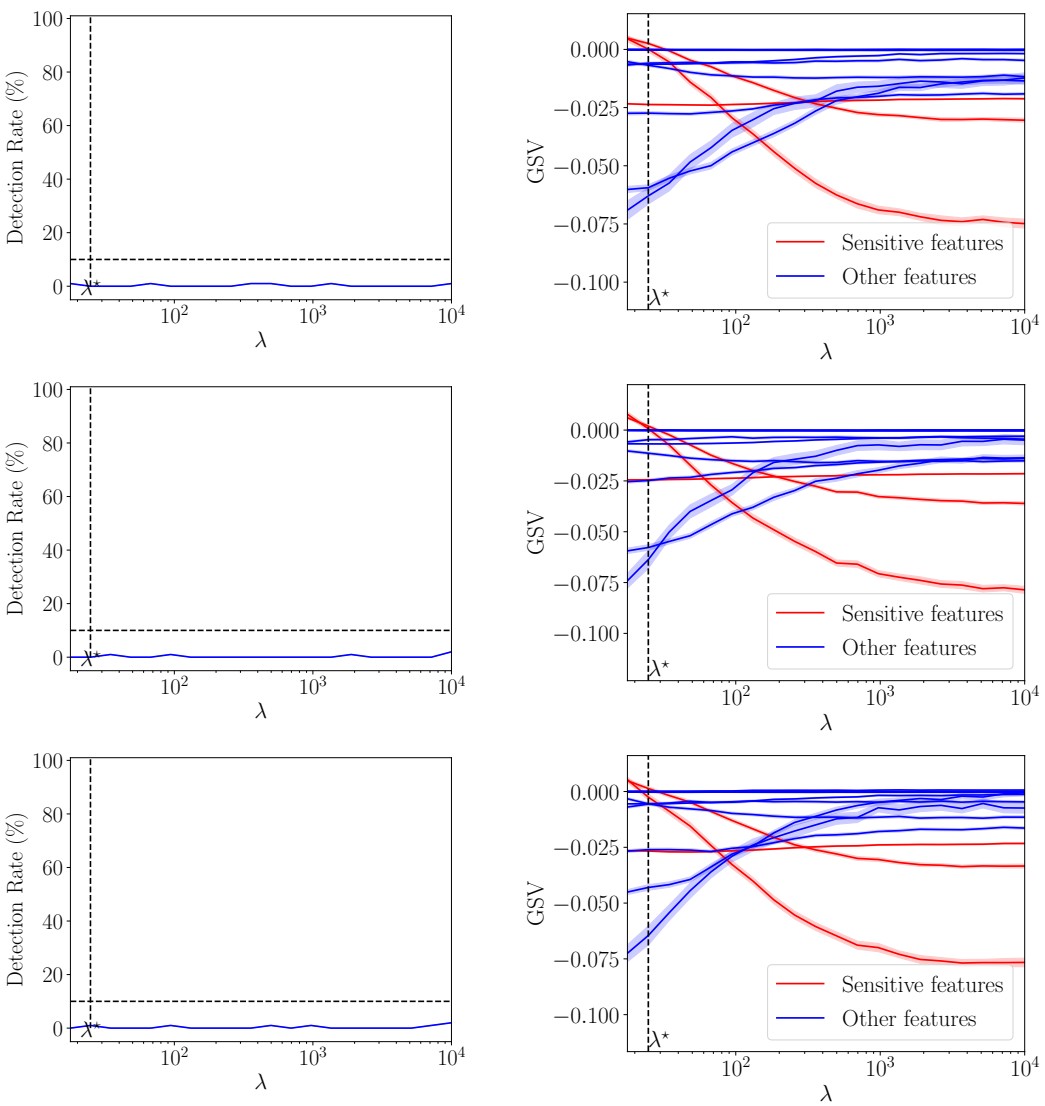

Figure 20: Example of log-space search over values of $\lambda$ using RFs classifier fitted on Adults and three sensitive attributes. Each row is a different train/test split seed. (Left) The detection rate as a function of the parameter $\lambda$ of the attack. (Right) For each value of $\lambda$, the vertical slice of the 11 curves is the GSV obtained with the resulting $\mathcal{B}'_{\boldsymbol{\omega}}$. The goal here is to reduce the amplitude all sensitive features (red curves) in order to hide their contribution to the disparity in model outcomes.

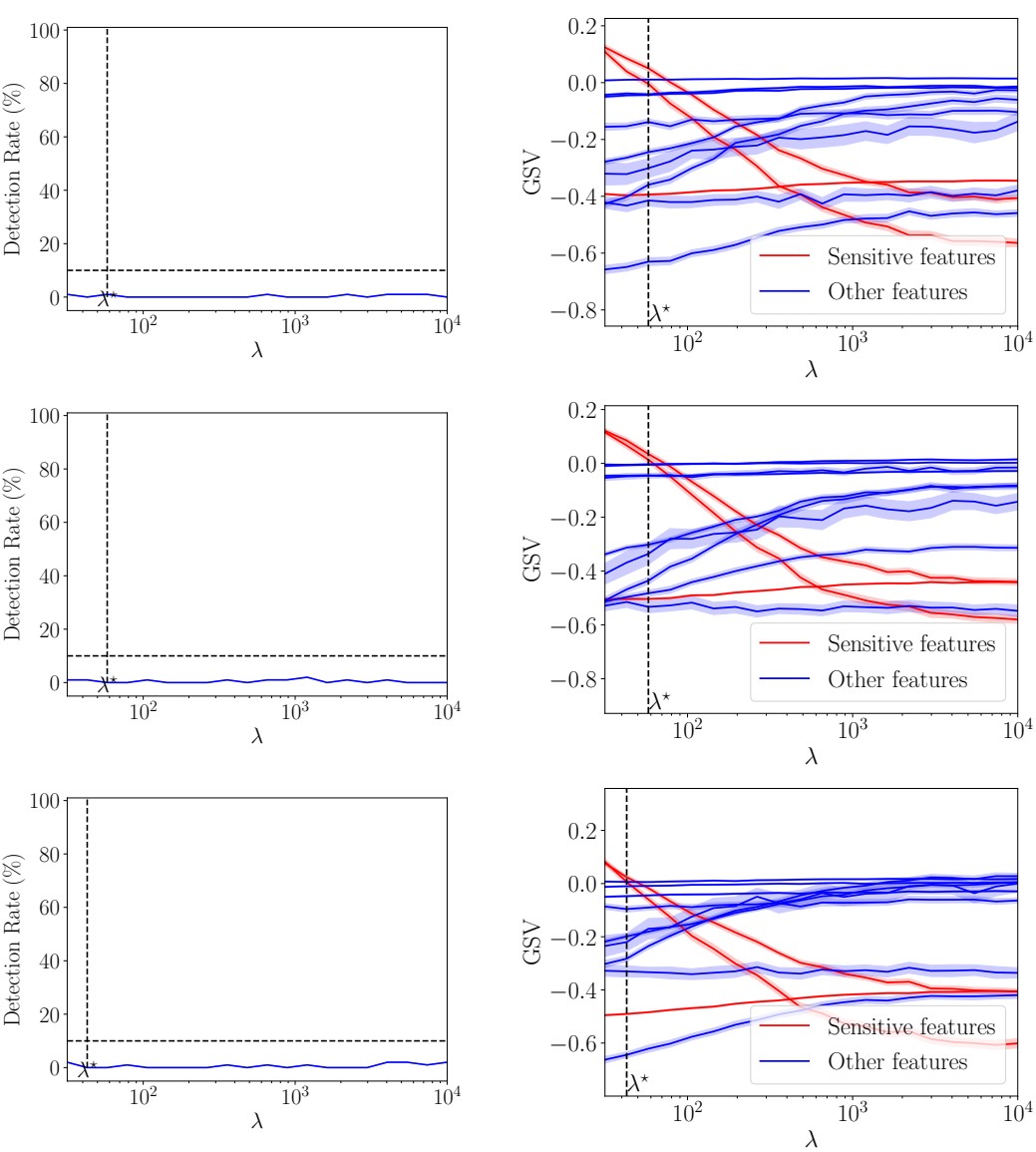

Figure 21: Example of log-space search over values of $\lambda$ using XGBs classifier fitted on Adults and three sensitive attributes. Each row is a different train/test split seed. (Left) The detection rate as a function of the parameter $\lambda$ of the attack. (Right) For each value of $\lambda$, the vertical slice of the 11 curves is the GSV obtained with the resulting $\mathcal{B}'_{\omega}$. The goal here is to reduce the amplitude all sensitive features (red curves) in order to hide their contribution to the disparity in model outcomes.

