# OpenReview forum: "Fooling SHAP with Stealthily Biased Sampling"
_ICLR.cc/2023/Conference — ICLR 2023 poster_

### Official Review · Reviewer_yVbY · 2022-10-24

**Confidence:** 4
**Correctness:** 3
**Technical Novelty And Significance:** 4
**Empirical Novelty And Significance:** 4
**Recommendation:** 8

**Clarity, Quality, Novelty And Reproducibility:**

The study is excellently positioning its standing in the prior work and is complete with clear conclusions together with ethics and reproducibility statements. Its algorithm explanations, derivations, and formalizations were clear, correct, and convincing. Its experiments used four real-world datasets and their use was convincingly justified. However, the authors could have described in the ethics statement also how they analyzed and evaluated that the datasets were created ethically and that their use of its for the purposes of this study was ethical. They could have also discussed in their ethics statement broader impact of their work because fairness is such a societally important consideration in machine/deep learning studies.

For further improving the clarity of this paper, please
* consider including numeric evidence (e.g., the number of datasets used in the evaluation) in the abstract to be more convincing,
* consider adding topic sentences to help the reader to understand the key message of each paragraph,
* consider if the flow of the paper could be improved by some re-organising to follow a more traditional IMRaD structure,
* check if some details (e.g., algorithms, examples, method descriptions) could be moved to the appendix to save some space from this part of the paper to strengthen the discussion part of the paper (the discussion topics were communicated in an outstanding way in the abstract but I would have wanted to read more about these take-home messages in the last sections of the study),
* consider if using past tense would work better in the experiments section, and
* proofread the paper for its punctuation once more (e.g., algorithms should be punctuated and check for comma/no comma with which).

**Details Of Ethics Concerns:**

To propose minor improvement here, the authors could have described in the ethics statement also how they analyzed and evaluated that the datasets were created ethically and that their use of its for the purposes of this study was ethical (please note that no new data was collected or released as part of this study). They could have also discussed in their ethics statement broader impact of their work because fairness is such a societally important consideration in machine/deep learning studies.

**Strength And Weaknesses:**

This is a carefully prepared, complete, and convincing paper in computer science about Shapley Values. It challenged the established statistics tool of SHAP that is yet to emerge as a widely adopted Interpretable AI method in evaluating machine/deep learning methods. Namely, the challenge was implemented by proposing "a complementary family of attacks that leave the model intact and manipulate SHAP explanations using stealthily biased sampling of the data points used to approximate expectations w.r.t the background distribution." Its results illustrated that SHAP can be fooled, thereby begging the further research questions on Shapley Values as a fair and reliable tool in evaluating, or even auditing, features in machine/deep learning applications.

See some minor comments related to improving the clarity of this paper below in the next section.

I would also encourage the authors to engage with a statistician (e.g., through research project support, statistical consultation unit, statistics services, or similar of their university) to ensure that their study design follows scholarly conventions in statistics. The paper excelled as a computer science study but I noticed that, for instance, the chosen significance level of 1% was somewhat unusual as typically \alpha = 0.005 or 0.05 is selected.

**Summary Of The Paper:**

This is a carefully prepared, complete, and convincing paper in computer science about Shapley Values. It challenged the established statistics tool of SHAP that is yet to emerge as a widely adopted Interpretable AI method in evaluating machine/deep learning methods. Namely, the challenge was implemented by proposing "a complementary family of attacks that leave the model intact and manipulate SHAP explanations using stealthily biased sampling of the data points used to approximate expectations w.r.t the background distribution." Its results illustrated that SHAP can be fooled, thereby begging the further research questions on Shapley Values as a fair and reliable tool in evaluating, or even auditing, features in machine/deep learning applications.

**Summary Of The Review:**

This is a timely and good paper and hence, I recommend its acceptation to ICLR 2023.

---

### Official Review · Reviewer_fFZy · 2022-10-30

**Confidence:** 4
**Correctness:** 4
**Technical Novelty And Significance:** 2
**Empirical Novelty And Significance:** 2
**Recommendation:** 6

**Clarity, Quality, Novelty And Reproducibility:**

My main concern is the choice of SHAP attribution method in this paper. There are two types of SHAP, observational vs. interventional( also called off manifold vs. on manifold), and it has been widely studied (e.g. [1- 3]) that interventional type of SHAP is not robust and can be counter-intuitive, because the computation is done on some data points that are not possible to exist (out of data manifold). In this paper, the attribution is done as the interventional type of SHAP,  which makes the manipuation not surprising given that there has been some acknowledgements on the nonrobustness of interventional SHAP. In interventional SHAP computations, the "absent" features are averaged using the same population distribution, together with the linearity, it takes the form of Equation (6), and makes the proof of Proposition4.1 convenient. However, in observational SHAP computations, "absent" features are averaged using conditional distribution, and it doesn't take the form of Equation (6), which makes Proposition 4.1 not easy to migrate. If the authors can show that the observational SHAP is also vulnerable to similar attacks, it would be very interesting, and as far as I konw would be the first work to show the nonrobustness of observational SHAP. However, the nonrobustness of interventional SHAP is not new.

Another concern of mine is the experiments. While I agree that the experiments show the effectiveness of the manipulation. I wonder how brute-force manipulation compares with the proposed stealthily sampling? By brute-force manipulation, I mean the company can sample S_1' honestly using uniform distribution, but repeatedly doing so and choose the one they like the most (non detectable and has the smallest attribution on the sensitive feature). It would be nice to see if the proposed algorithm has a significant improvement over the brute-force method, controlled for time complexity.

Finally I have a question for clarification as I am not very familiar with the auditing problem. On page 4, what do you mean exactly by "the model is explicitly relying on x_s"? How does one measure the "explicitness"? What if the model relies on a different but almost redundent feature to the sensitive feature? If SHAP is used, I imagine one can add many more features almost redundent to the sensitive / audited feature, this will decrease the sensitive feature's attribution at the speed of 1/k, if k more features are added. This would work in both observational and interventional SHAP, but of course it would be manipulating model rather than manipulating samples.

[1] Shapley explainability on the data manifold
[2] Problems with Shapley-value-based explanations as feature importance measures


**Strength And Weaknesses:**

Pros:
The paper's presentation is very clear, the introduction of the auditing scenario is well motivated and interesting. The paper, as well as the supplemental code, is well organized and easy to follow.

Cons:
The vulnerability of the interventional SHAP (more details below) is not entirely new. While the experiments do show the effectiveness of the proposed manipulation algorithm, its superiority over brute-force/baseline method is not provided theoretically or experimentally.

**Summary Of The Paper:**

This paper studies the manupulation of the feature attributions method, so that the attributino to sensitive features are hidden. More concretely, this paper studies the audit scenario, where the auditors are given a full set of all predictions, and a small sample of features; the authors then propose an algorithm (called stealthily biased sampling) to get the small samples to submit to audit, so that 1. the auditors cannot detect the manipulation by studying the prediction distribution, and 2. the attribution to some sensitive feature given by SHAP is decreased.

**Summary Of The Review:**

Overall I think the paper is very well written, and studies an interesting auditing scenario in details. However, this vulnerability to the manipulation is not entirely new. I give 5 for now, but will reconsider if my concerns are addressed.

---

### Official Review · Reviewer_L3wE · 2022-11-10

**Confidence:** 4
**Correctness:** 4
**Technical Novelty And Significance:** 2
**Empirical Novelty And Significance:** 2
**Recommendation:** 6

**Clarity, Quality, Novelty And Reproducibility:**

Clarity:
The paper is well written and the argument is easy to follow.

Quality:
The work sets a modest goal, and achieves it quite convincingly.

Novelty:
To me this is the main point. I guess the paper marries Baniecki & Biecek (2022) idea of manipulating
SHAP with Fukuchi et al. (2020) attack. I think this paper is arguably written better (crisper) than Fukuchi,
but I'm not sure what really  is new.
I  will commend the authors on being very straight-forward about their contributions.

I saw no reproducibility problems in the paper.

**Strength And Weaknesses:**

Strength:
The paper is well organized, very clear in its communication, and displays well the results.
I found it to be a very nice read.

Weakness:
The main weakness is that it is not clear how different this paper is from Fukuchi et al.
It seems that both the attack, and more generally almost all the methodology (Wasserstein distance) can already be found there.
Moving this idea to SHAP is nice, but the question is whether there was anything interesting to say about this move.
In other words:
- I do not understand whether there are any substantial changes that the method  discussed in   Fukuchi et al.
(2020) had to undergo to work for the SHAP values.
In a sense, the  SHAP values themselves are almost not discussed at all  in the paper; do they provide different bounds?
did adapting Fukuchi et al.
(2020) required any extra work, or indeed if the method works differently in some sense when adapting it to SHAP.

**Summary Of The Paper:**

The paper discusses the use of contrastive Shapley values as a way of post-hoc auditing of prediction models.

The scenario that an auditor asks for samples from two populations according to a sensitive demographic feature, which should not appear in the model. Then, the auditor checks the distribution of Shapley values to see if there are positive or negative differences across these populations.

The paper argues that by making small changes to the sampling weights, the company being audited can attack the audit and alter it's results. The paper provides an optimization score that weighs the changes in the distribution (in Wasserstein distace) against the change in the global Shapley values, and an algorithm for minimizing this optimization problem.
They show that cosiderable changes to the audit score can be achieved while still being hard to detect.

In terms of the main cotribution, as outlined in section 4.4,  it is moving the idea of stealth sampling from the original important feature to the SHAP value computed for the feature.


Update: I'm changing my evaluation to marginally accept. Again, I think this paper is written clearly and is therefore useful in presenting the ideas. My main concern is the amount of novelty required for ICML. The answers to my questions have been clear about the nature of the  novel contributions. I think at this point this an editorial call that I am happy to leave to the area-chairs etc.



**Summary Of The Review:**


Paper is written very nicely.
It's not clear to me that the changes from previous methods are sufficient for acceptance.


PS. I appologize for this very late review.

---

### Author Response · Authors · 2022-11-09
**Official Comments by the Authors**

We wish to thank both reviewers for their insightful feedback and comments. We have updated the paper in light of your comments and have highlighted the changes in blue. Here are some of the major changes
- Table 1 presents the false positive rates of the detector using a 5% significance level, which is more standard in statistics.
- Figure 5 in the result sections now presents the relative decrease in the attribution of the sensitive attribute. Fool SHAP (our method) and the brute-force approach proposed by Reviewer fFZy are compared. To make meaningful comparisons between the two attacks, the brute-force approach was only allowed to run for the same amount of time it took to compute the non-uniform weights with Fool SHAP (typically 30-180 seconds)
- The discussion sections has been completely reworked.
- The abstract now contains numerical results.

---

### Author Response · Authors · 2022-11-14
**Second Official Comments by the Authors**

In light of the new review, we have updated the document once more. Here are the two key changes
- We have changed section 4.4 to better highlight our technical contributions relative to the genetic algorithm of Baniecki & Biecek (2022)
and the Stealth framework of Fukuchi et al. (2020).
- We present new results in appendix E.3 on the performance of the genetic algorithm of Baniecki & Biecek. We note that the audit is able to easily detect the manipulation of the subset $S_1'$ because the genetic algorithm does not take into account the fact that $S_1'$ should be similar in any way to $D_1$. We intend to present more results on this subject by the end of the rebuttal period.

---

### Decision · Program_Chairs · 2023-01-20

**Decision:**

Accept: poster

**Justification For Why Not Higher Score:**

The negatives of the paper are that it simply ports stealth sampling from the feature to the Shapley value and that it focuses on an interventional Shapley value that evaluates models out of support.

**Justification For Why Not Lower Score:**

The positives of this paper are that it is clearly written and points out a problem in a variant of Shapley values.

**Metareview: Summary, Strengths And Weaknesses:**

This paper demonstrates another angle by which a certain Shapley can be manipulated. The new angle centers on biased sampling rather than model editing. The paper studies the technique in a fairness audit, where they show a large decrease in amplitude of the sensitive feature. The positives of this paper are that it is clearly written and points out a problem in a variant of Shapley values. The negatives of the paper are that it simply ports stealth sampling from the feature to the Shapley value and that it focuses on an interventional Shapley value that evaluates models out of support. Regarding the last point see also

@inproceedings{jethani2021fastshap,
  title={FastSHAP: Real-Time Shapley Value Estimation},
  author={Jethani, Neil and Sudarshan, Mukund and Covert, Ian Connick and Lee, Su-In and Ranganath, Rajesh},
  booktitle={International Conference on Learning Representations},
  year={2021}
}

who use a surrogate model to keep things "in-distribution."

The reviewers are now all positive. Two reviewers have increased their score given the author reply.

**Note From Pc:**

if the above contains the word "oral" or "spotlight" please see: "oral" presentation means -> notable-top-5% and "spotlight" means -> notable-top-25%. As stated in our emails, we are disassociating presentation type from AC recommendations